# JNK signaling in pioneer neurons organizes ventral nerve cord architecture in *Drosophila* embryos

Katerina Karkali[1,2,3], Timothy E. Saunders [2,4], George Panayotou[3] & Enrique Martín-Blanco [1,2] ✉

Morphogenesis of the Central Nervous System (CNS) is a complex process that obeys precise architectural rules. Yet, the mechanisms dictating these rules remain unknown. Analyzing morphogenesis of the *Drosophila* embryo Ventral Nerve Cord (VNC), we observe that a tight control of JNK signaling is essential for attaining the final VNC architecture. JNK signaling in a specific subset of pioneer neurons autonomously regulates the expression of Fasciclin 2 (Fas 2) and Neurexin IV (Nrx IV) adhesion molecules, probably via the transcription factor *zfh1*. Interfering at any step in this cascade affects fasciculation along pioneer axons, leading to secondary cumulative scaffolding defects during the structural organization of the axonal network. The global disorder of architectural landmarks ultimately influences nervous system condensation. In summary, our data point to JNK signaling in a subset of pioneer neurons as a key element underpinning VNC architecture, revealing critical milestones on the mechanism of control of its structural organization.

The complex nervous system of bilaterians is thought to have emerged from an ancient, diffuse nerve plexus[1], which evolved into a stereotyped, segmented, polarized structure[2]. Within this framework, functional neural circuits[3–5] developed from precise 3D arrangements of neurons and glial cells[6]. Neuro-glial interactions and axon guidance decisions are directly coupled to the nervous system's architecture inception. Yet, how the nervous system achieves a robust mechanically-balanced organization throughout morphogenesis, while neuronal connectivity is being established, is poorly understood. To explore this challenge, we made use of the developing *Drosophila* embryo Central Nervous System (CNS), in particular the Ventral Nerve Cord (VNC).

The mature VNC is composed of segmentally iterated ganglia with a stereotyped pattern of neurons organized in a 3D array. Neurons are surrounded by a cellular meshwork composed of different types of glia and are linked by cohesive axon bundles that establish longitudinal connectives and transverse commissures along the neuropile[7]. The neuronal cell bodies in the VNC are restricted to the cortex, while sensory axons and interneuron projections mostly accumulate in more medial domains. The positions of neurons and their projections have been mapped by employing segmentally repeated reference points[8]. As embryonic development proceeds, axonal and dendritic arborizations become more complex while the VNC condenses along the anterior-posterior (AP) axis. Condensation is bidirectional and oscillatory and responds to the coordinated contractile activities of neurons and glia[9].

Although the signaling mechanisms directing axon guidance are well understood[10], those acting on the structural organization of the nervous system remain less explored. Considering its role as a stress response element[11] and its capacity to increase developmental robustness[12], the JNK cascade stands as a potential mediator on the mechanical control of the structural organization of the nervous system. Further, JNK recently emerged as a regulator for axonal growth and regeneration[13,14] conserved throughout evolution and as a key element in regulating glia in vertebrates[15]. *Jnk1;Jnk2* mice fail to complete neural

[1]Instituto de Biología Molecular de Barcelona (CSIC), Parc Cientific de Barcelona, Baldiri Reixac 10-12, 08028 Barcelona, Spain. [2]Mechanobiology Institute and Department of Biological Sciences, 5 Engineering Drive 1, National University of Singapore, Singapore 117411, Singapore. [3]BSRC Alexander Fleming, 34 Fleming Street, 16672 Vari, Greece. [4]Division of Biomedical Sciences, Warwick Medical School, University of Warwick, Coventry CV4 7AL, UK. ✉e-mail: embbmc@ibmb.csic.es

tube closure, displaying exencephaly[16]. In *Drosophila*, JNK (*basket - bsk*) participates in controlling axonal growth[13]. With these considerations, we hypothesized that the JNK pathway plays a regulatory role in the morphogenesis and structural organization of the VNC.

To test this hypothesis, we interfered with the pathway by modulating JNK kinase activity in early *Drosophila* embryos. This allowed us to uncover essential functions of the JNK pathway in orchestrating the structural pattern and condensation of the VNC. We find that early in embryogenesis, JNK signaling modulates pioneering events by regulating expression of cell adhesion molecules. This is achieved through the transcription factor Zfh1. Further, JNK signaling ensures the correct spatial arrangement and fasciculation of axons that facilitate VNC condensation. Altogether, our data uncover a fundamental contribution of the JNK signaling pathway in a subset of pioneer neurons to implement CNS architectural organization.

## Results

### Quantitatively defining the 3D structural elements of the Ventral Nerve Cord

We first asked which are the key elements, from an architectural point of view, sustaining the highly conserved structural pattern of the VNC. Amongst the most prominent structural components of the VNC are the longitudinal connectives, which are built by fasciculation of axons of different origins. In the *Drosophila* embryo a subset of these axons, running along the medio-lateral (ML) and dorso-ventral (DV) axis of the VNC, are immunoreactive for Fasciclin 2 (Fas 2), a neural cell adhesion molecule whose loss-of-function leads to complete or partial bundle defasciculation[17]. To investigate the potential structural role of Fas 2 positive fascicles, we performed a spatial 3-dimensional (3D) correlation analysis to detect conserved structural domains along the antero-posterior (AP) axis (Fig. 1a, b, see also Supplementary Movie 1). We compared in sequential transverse sections the positions of the Fas 2 positive axonal bundles. Contrary to the presumed even axial distribution of Fas 2 positive fascicles[8], the correlation analyses showed a specific, segmentally repeated, pattern of single structural domains displaying robust 3D geometrical organization. We refer to these regions as structural 3D nodes. At late stage 16 embryos, nodes distribute at an average distance of 28 μm along the AP axis (Fig. 1c), and in anatomical terms, stand slightly posterior to the segmental posterior commissures. We confirmed this distribution by double staining with well-known axonal scaffold markers (BP102 and 22C10) (Supplementary Fig. 1). Longitudinal connectives within these domains are organized in three compact fasciculated units with precise dorso-ventral / medio-lateral topologies (medial, intermediate and lateral) (Fig. 1d). Between the 3D nodes, Fas 2 axons spatially disperse from segment to segment and the noise in the correlation between successive domains (relative difference in the maximum and minimum spatial derivatives of aligned profiles) drastically increases (Fig. 1e). The correlation analyses also uncover a segmentally iterated structural bias along the AP axis. From the 3D nodes, correlation noise increases steeply in the posterior direction towards a medial inter-commissural position. Then, the accuracy of the correlation between domains recovers in a shallow gradient. In brief, the internodal domain is, from anterior to posterior, stereotypically asymmetric (Fig. 1f).

The 3D organization of axonal tracts and neuronal bodies dictates the size and shape of the *Drosophila* embryonic VNC. Yet, the spatial distribution of these elements, although segmentally iterated, is irregular and noisy. This prevents the interpretation of the mechanical processes shaping VNC architecture. Our correlation analyses of the Fas 2 positive connectives overcome this problem by revealing a quantifiable polarized 3D structural pattern for the VNC.

### VNC architecture is modulated by JNK signaling

JNK in *Drosophila* is essentially ubiquitous and its activity is mostly modulated by upstream activating kinases and regulatory phosphatases. In most *Drosophila* tissues, JNK activity is restrained by a negative feedback loop implemented by the dual-specificity phosphatase Puckered (Puc). *puc* expression depends on the activity of Bsk, the *Drosophila* homologue of JNK, and Bsk function is inhibited by Puc[18]. Although *puc* expression has been reported in the *Drosophila* embryonic CNS[18], its role, if any, in CNS morphogenesis is unknown.

We found that in *puc*[E69] and *puc*[B48] loss-of-function alleles, generated by independent transposon insertions in two different introns[19], the structural organization of the VNC was disturbed (Fig. 1g). In both cases, Fas 2 positive longitudinal connectives became highly defasciculated, with the most lateral ones being incompletely formed. In addition, the intersegmental nerves were unevenly distributed. Axons came together irregularly into more medial positions and an increase of Fas 2 signal was observed within the cell bodies (particularly in the aCC and pCC neurons) (Fig. 1g, Supplementary Fig. 2 and Supplementary Movie 2). Although the excess of JNK activity affected axonal organization, in both the DV and ML axes (Fig. 1h), the structural 3D nodes were formed. Interestingly, in *puc* mutants, the average internodal distance was increased (33 μm) (Fig. 1i and Supplementary Fig. 2) and internodal domains lost their iterated AP axial polarity (Fig. 1f). Accordingly, VNC condensation failed (on average by 18% in *puc*[E69] and by 39% in *puc*[B48] alleles) (see direct measurements of the VNC length in Supplementary Fig. 3). In conclusion, Puc activity seems to be involved in the structural reinforcement and condensation of the VNC, without influencing the outlining of the segmental body plan.

We then asked if defects in fasciculation and VNC condensation in *puc* mutants were a consequence of increased JNK activity. In the absence of the negative feedback loop mediated by *puc*, both in *puc*[E69] and *puc*[B48] embryos, the level of JNK phosphorylation significantly rose in the developing VNC, re-locating in part to cell bodies (Supplementary Fig. 4). This strongly supports the existence in the CNS, as previously described in other developmental processes[19], of negative feedback on JNK activity mediated by *puc*. The increase in JNK activity was also associated to an increase in apoptosis over the basal cell death levels characteristic of the developing VNC (Supplementary Fig. 4).

In summary, a tight control of JNK signaling activity appears to be essential for the acquisition of the proper structural organization and architectural balance of the developing VNC, and for its condensation. This control is exerted, at least in part, by Puc, which restricts JNK activity and also sustains cell viability.

### JNK signaling is dynamically regulated in the VNC

To determine when and where the JNK cascade is active in the VNC, we studied the dynamics of *puc* expression, which is itself an early readout of the pathway[19]. In heterozygosity, a viable Gal4 line under the control of the *puc*[E69] enhancer (*w1118; UAS-GFP; puc*[E69]*I–Gal4 / TM6B*)[20] was expressed in a segmentally iterated subset of neurons (Fig. 2a, b), without cross-reacting with glial markers[21] (Fig. 2c, d). At stage 13-14, *puc* expression initiated in a few neurons at the ventral midline (Fig. 2e and Supplementary Movie 3). This expanded to a minimum consensus of 18 *puc* positive neurons per hemisegment at late embryonic stage 16 (see representative diagrams in Fig. 2f and Supplementary Movie 4).

Dual staining of *puc*[E69] embryos with neuronal markers (Even-skipped (Eve) and Engrailed (En)) enabled the identification of most *puc* positive cells (Supplementary Fig. 5). *puc* labeled the aCC and RP2 intersegmental nerve (ISN) pioneers[22], the U1-U3 motoneurons (plus weak/variable expression in U4-U5), as well as the longitudinal connectives pioneer, pCC[23], and 3 out of the 8 to 10 EL interneurons. Meanwhile, NL1, NL2, and iVUM5 En positive neurons were also *puc* positive. Morphological and positional criteria further enabled the assignment of *puc* expression to 5 Eve / En negative neurons:, 2 VUM motoneurons; the AD and ADV neurons; and the longitudinal connectives pioneer dMP2[23] (Fig. 2g). It is relevant to point that not all pioneers express *puc* and that at early stage 13, *puc* expression was limited to a subset of the Fas 2 positive cells. The longitudinal

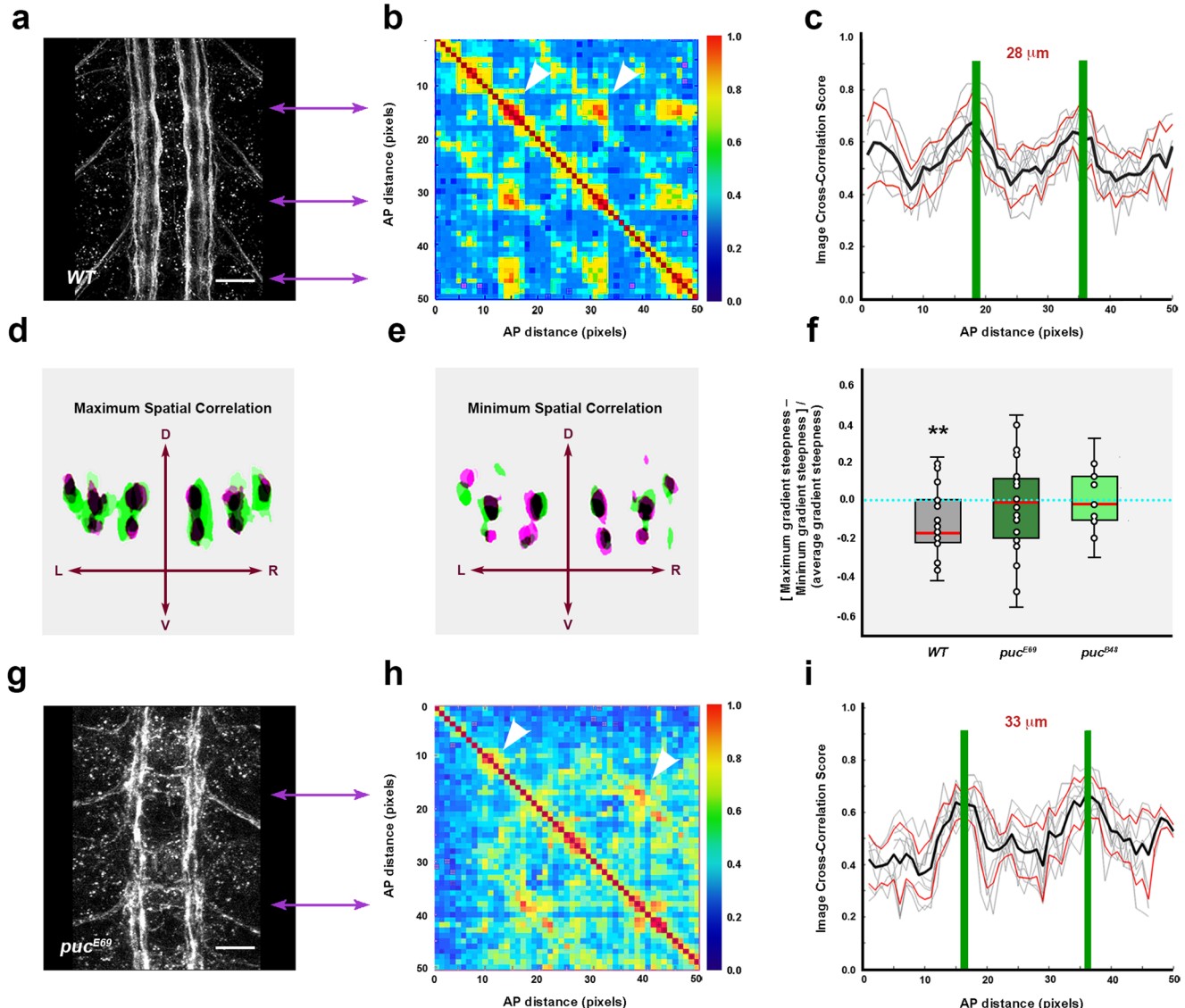

**Fig. 1 | Structural Architecture of the Ventral Nerve Cord (VNC) of the *Drosophila* embryo. a** Maximum projection of the VNC of a stage 16 wild type embryo stained with Fas 2. Anterior is up. See Supplementary Movie 1. Scale Bar is 10 μm. **b** Self-cross-correlation matrix of the Z sections of the image in (**a**) along the AP axis (in pixels). The cross-correlation matrices correspond to hemisegments. The panel is scaled to fit with (**a**). The pixel size after the scaling is 1.65 μm. The color-coded representation (0–1), in all figures from now on, shows the correlation level along the AP axis. One conspicuous node per segment was detected (white arrowheads). Double headed purple arrows point to 3D nodes in (**a**). **c** Image cross-correlation score along the AP axis for (**b**) (*n* = 6). Fas 2 3D nodes fall 28 μm apart and the internodal minimum correlation maps 9 μm posterior to the node. The grey lines show individual profiles. Black line: average values; red lines ± SD; green bars indicate maximum correlation (3D nodes). **d** Z sections at two different maximal (green and magenta) correlation positions (from (**a**)). DV and ML [Left (L) and

Right (R)] axes are indicated by arrows. **e** Z sections at two different minimal (green and magenta) correlation positions (from (**a**)). DV and ML axes are indicated by arrows. **f** Gradient of spatial correlation along the AP axis. Data was derived from *n* = 3 embryos, in each condition, with at least 3 positions compared. Red bar denotes the median. Top and bottom of the boxes indicate the 25th and 75th percentiles. Whiskers extend to the maximum and minimum values. Mann–Whitney tests were employed. Statistically significant differences from 0 were detected for the wild type (**p* = 0.008). **g** Equivalent image to (**a**) for a *puc^E69^* embryo. Note the defasciculation and collapse of the longitudinal connectives. See Supplementary Movie 2. Scale Bar is 10 μm. **h** Self-cross-correlation matrix of the Z sections of (**g**). In *puc^E69^*, matrices are noisier although a single node per segment persists (white arrowheads). **i** Image cross-correlation score along the AP axis for (**h**) (*n* = 9). Fas 2 3D nodes spread 33 μm away and the internodal minima fades out. Lines and bars are coded as in (**c**). Source data are provided as a Source Data file.

connectives pioneers vMP2 and MP1 do not express it (Fig. 2h). A few stochastically distributed *puc* positive neurons, which do not form part of our consensus set, were also occasionally observed. These neurons presented very low GFP levels or variable expressivity and were not considered further. Importantly, the pattern of expression of *puc^E69^* was largely conserved for the *puc^B48^* enhancer (*w1118; UAS-GFP; puc^B48^–Gal4 / TM6B*) (Supplementary Fig. 6).

In summary, high levels of *puc* expression, most probably denoting high preceding JNK activity, were detected in VUMs, pCC and possibly dMP2 neurons at the ventral midline from stage 13. This

expression pattern evolved to include diverse inter and motoneurons, the axons of which fasciculate to the longitudinal connectives, and the segmental and intersegmental nerves.

## The level of JNK activity in *puc* positive neurons shapes VNC architectural organization

The decrease in VNC architectural robustness observed in *puc* could be the result of autonomous aberrant activation of the JNK pathway in the CNS, or just a secondary structural consequence of the general disruption of the embryo anatomy. To rule out this latter possibility, we

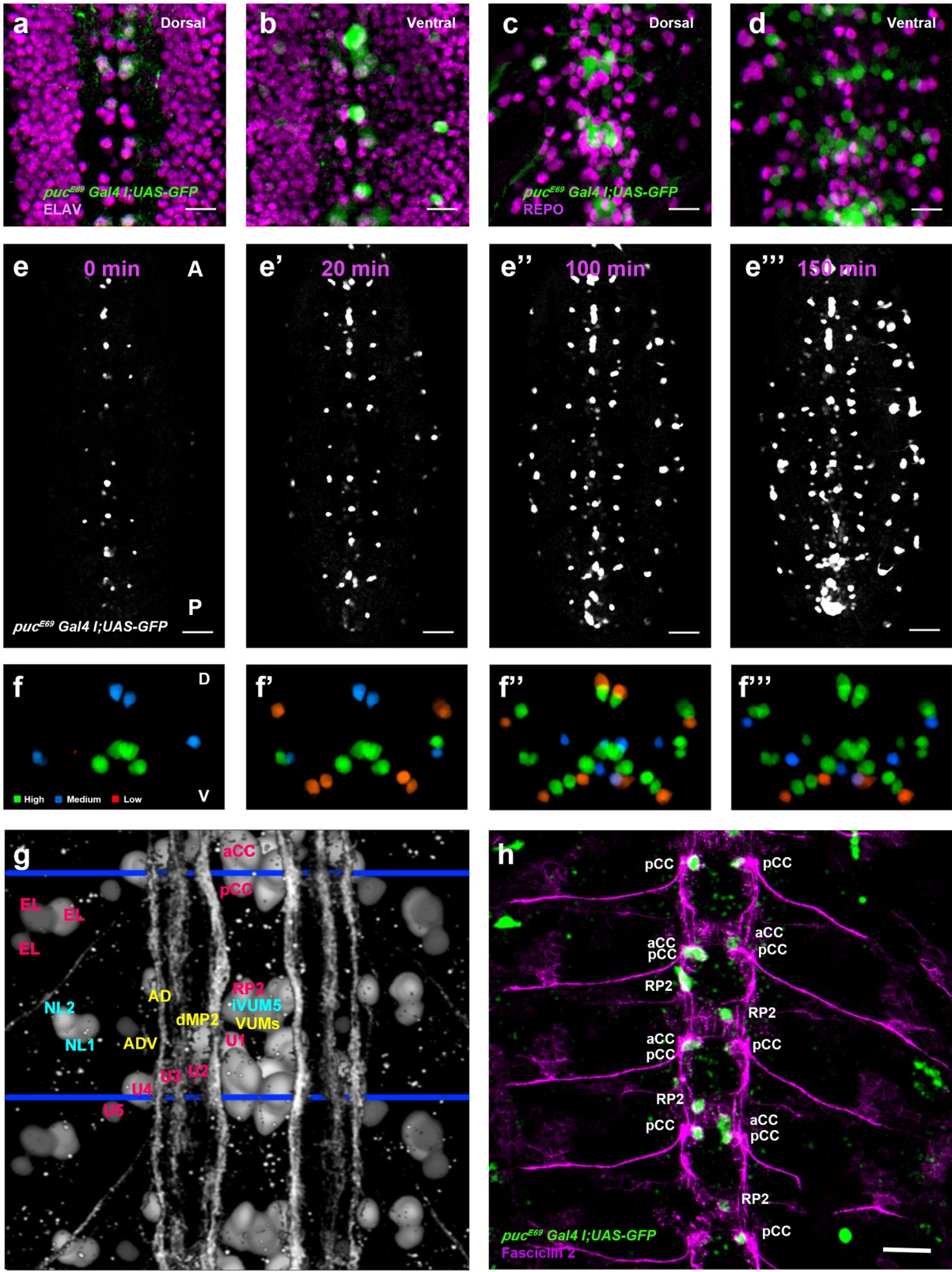

employed the dual Gal4/UAS system to generate JNK signaling loss- or gain-of-function conditions overexpressing either Bsk[DN], a dominant negative form of Bsk, or Hep[CA], a constitutively active form of Hemipterous (Hep), the *Drosophila* homologue of the JNK kinase MKK7. We targeted specific subsets of the *puc*-expressing cells employing different Gal4 lines (CQ2, RN2 and mzVUM).

The *RN2-Gal4* line is specifically expressed in the *puc* positive aCC and RP2 motoneuron ISN pioneers and the pCC longitudinal connective pioneer (Fig. 3a). Suppressing JNK activity in RN2 positive cells resulted in the disorganization of the Fas 2 positive connectives (Fig. 3b). Intersegmental correlations became noisy and irregular, internodal distances were enlarged, and the condensation of the VNC

**Fig. 2 | JNK signaling is dynamically regulated during development. a–d** Dorsal and ventral sections highlighting the expression of *puc* (green) in the VNC of stage 17 *puc$^{E69}$I-Gal4> UAS-GFP* embryo, double-stained for the pan-neuronal marker Elav ((**a**) and (**b**) - magenta) or the glial marker Repo ((**c**) and (**d**)- magenta). *puc* is expressed in a subset of neurons in each segment but not in glia. Scale bar 10 μm. **e–e′′′**) four different snapshots from Supplementary Movie 3 at the indicated time points from the *UAS-GFP; puc$^{E69}$I-Gal4* viable line. *puc* expression can be observed from stage 14 in a few neurons at the midline. As development progresses, further neurons express *puc*. Scale bar 50 μm. **f–f′′′** Diagrams of cumulative *puc* expression at the same time points as in **e–e′′′**. A minimum consensus of 18 neurons per hemisegment was defined. At each time point the intensity (defined by visual inspection) was displayed as low (red), medium (blue) and high (green) (see also Supplementary Movie 4). **g** Composition summarizing the expression of *puc* in the CNS [*puc* expressing cells in the VNC, at stage 16, including those identified by coexpressing Eve (magenta) or En (cyan) plus those putatively identified by morphological and positional criteria (yellow - AD, ADV, dMP2 and 2 VUM En-negative motor neurons)]. To facilitate understanding, a scaled Fas 2 expression image from an embryo of the same stage was over-imposed to the diagram. Positions of the 3D nodes are highlighted by dark blue lines. **h** Maximum projection of the VNC of a stage 13 *puc$^{E69}$I-Gal4> UAS-GFP* embryo double-stained with Fas 2 (Magenta). The Fas 2 positive pioneers expressing *puc* are indicated. Note that GFP expression penetrance at this stage is partial. Anterior is up. Scale bar 20 μm.

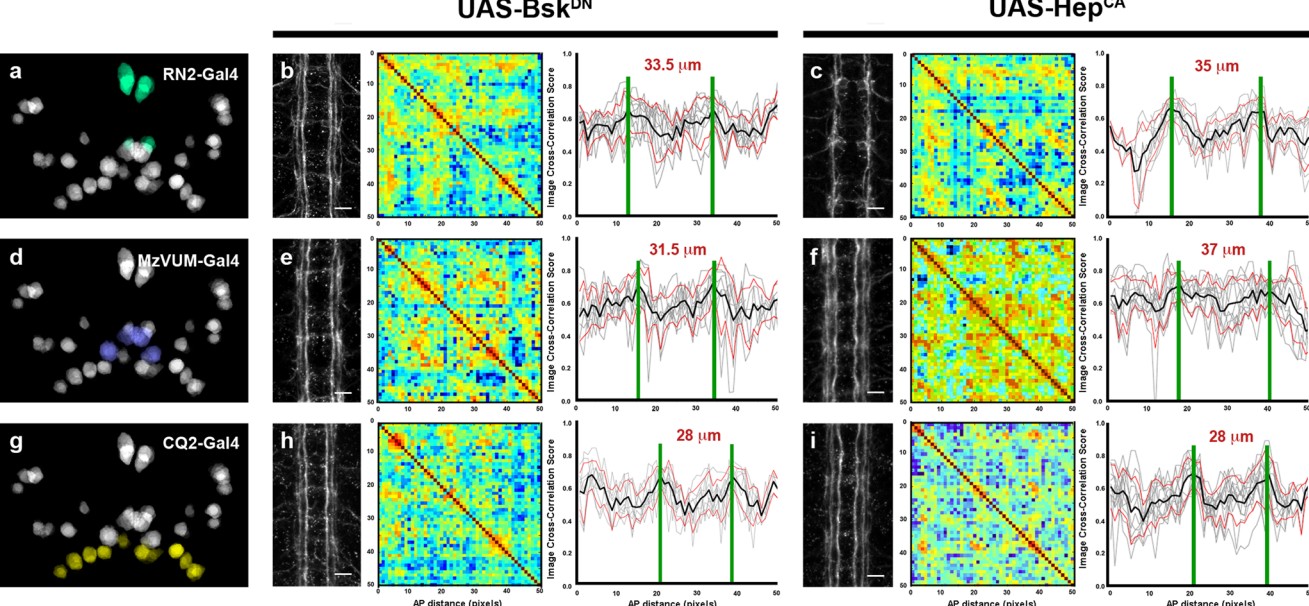

**Fig. 3 | Targeted activation and inhibition of the JNK pathway in the VNC. a** Cartoon representing the full set of *puc* expressing neurons per segment in the VNC, highlighting cells expressing *RN2-Gal4* (turquoise). From left to right, pattern of Fas 2 expression, self-cross-correlation matrix and image cross-correlation profiles (scales and colormap as in Fig. 1) for VNCs of 16-stage embryos upon expression of *UAS-Bsk$^{DN}$* **b** and *UAS-Hep$^{CA}$* **c** transgenes, under the control of the *RN2-Gal4* respectively, inhibiting and hyperactivating JNK activity in the aCC, pCC and RP2 cells. In this condition connectives defasciculation and 3D nodes cross-correlation profiles are affected. Grey lines show individual profiles. Black line corresponds to average, with red lines ± SD. Green bars indicate positions of maximum correlation (3D nodes) along the AP axis. Distances between nodes are indicated. Scale bar 10 μm. **d** Cartoon highlighting the *puc* + neurons expressing *MzVum-Gal4* (purple). **e, f** As (**b**) and (**c**) but employing *MzVum-Gal4* for interfering in JNK activity in the VUM neurons. Affecting JNK activity levels in VUMs also affects defasciculation, 3D nodes profiles and condensation. Scale bar 10 μm. **g** Cartoon highlighting the *puc* + cells expressing *CQ2-Gal4* (yellow). **h and i** As (**b**) and (**c**) but employing *CQ2-Gal4* for interfering in JNK activity in the U motoneurons. This condition does not have a phenotype distinct from wild type. Scale bar 10 μm. Control data are presented in Fig. 1. Micrographs and self-cross-correlation matrices correspond to representative embryos for each condition. Image cross-correlation profiles present the cumulative data (mean ± SD) of all embryos analyzed ((**b**), *n* = 10; (**c**), *n* = 10; (**e**), *n* = 9; (**f**), *n* = 10; (**h**), *n* = 10; (**i**), *n* = 10).

failed. Conversely, hyperactivation of the pathway, mimicking reduced *puc* levels, also strongly affected Fas 2 connectives, which kept a juvenile appearance (Fig. 3c). Although robust structural 3D nodes were sustained, the internodal distance enlarged and the VNC lengthened.

The *MzVum-Gal4* line is expressed in a few lateral cells and in the three VUM motoneurons (Fig. 3d). Bsk$^{DN}$ expression slightly affected fasciculation of Fas 2 connectives (Fig. 3e) but did not result in overt alterations in intersegmental correlations or internodal profiles. Conversely, Hep$^{CA}$ expression resulted in disorganization of Fas 2 connectives and increased noise in the spatial correlations (Fig. 3f). Internodal distances became larger and localized spatial information was mostly lost.

The *CQ2-Gal4* line drives expression of UAS-linked genes in the U1 to U5 motor neurons, a subset of Eve-positive neurons, which also express *puc* (Fig. 3g). Neither the inhibition (Fig. 3h), nor the hyperactivation of the pathway (Fig. 3i) resulted in any major alteration of the Fas 2 expressing connectives. The VNC architecture in both cases

(*e.g.*, node position and spatial correlation profiles) was similar to wild type.

Altogether, these data indicate that both down- and hyper-regulation of the JNK pathway in early-specified aCC, pCC, and RP2 pioneers and in the VUM neurons, led to global architectural defects. While these defects were more evident at the lateral Fas 2 bundle, which is the last to be completed and hence likely the most sensitive to perturbation, the correlation analyses showed that the whole 3D organization of the Fas 2 axonal network was disrupted. The non-identical, but similar defects observed upon Bsk$^{DN}$ and Hep$^{CA}$ over-expression in these cells underscore two relevant matters; 1) the overall structural organization of the VNC seems to require strict local control of the activity of the JNK signaling cascade, and 2) the level of activity of the pathway appears to be subjected to a negative feedback regulatory loop that depends on the transcriptional control of the dual specificity phosphatase Puc. In agreement, in *puc* positive neurons (*puc$^{E69}$I-Gal4* pattern) at stage 16, *puc* expression was downregulated in response to both the hyperactivation (Hep$^{CA}$) and the downregulation

(Bsk$^{DN}$) of the pathway (Supplementary Fig. 7). This is not without precedents[19,24–26]. Bsk$^{DN}$ directly interferes with endogenous JNK activation, reducing *puc* expression, while Hep$^{CA}$ initially boosts Puc production, which will shut off Bsk, leading to *puc* downregulation in the long term. In both conditions, most probably as a result of temporal delays and compensatory effects, VNC length was unaffected (*puc$^{E69}$I-Gal4> UAS-GFP*, 303 ± 9 (*n* = 9); *puc$^{E69}$I-Gal4> UAS-GFP::UAS-Hep$^{CA}$*, 295 ± 15 (*n* = 8) and *puc$^{E69}$I-Gal4> UAS-GFP::UAS-Bsk$^{DN}$*, 294 ± 16 (*n* = 7)).

## JNK signaling in pioneer neurons modulates Fas 2 expression

Alterations in JNK activity in pioneer and VUM neurons non-autonomously affect the overall 3D organization of the VNC. While growth factors or neurotransmitters can be an important source of instructive information[27], adhesion molecules themselves (*e.g.*, Fas 2) may also act as key elements for communication in large-scale processes[28–31].

In this context, we explored whether the global effects of *puc* were associated with deregulation of cell adhesion components. Beyond gene regulation, it is well known that JNK signaling can modulate cytoskeleton dynamics and cell adhesion[32]. By quantifying fluorescence integrated densities in wild type, *puc$^{E69}$*, and *puc$^{B48}$* embryos immunostained for Fas 2 (Fig. 4a–c), we found that Fas 2 levels were not uniform along the axonal connectives, neither in wild type animals nor in the different *puc* alleles. Fas 2 expression was enriched at the structural 3D nodes (Fig. 4d). In *puc* mutant conditions, Fas 2 expression was consistently 50% lower than wild type. This was corroborated by immunoblotting in total extracts of stage 16 embryos (Fig. 4e). These analyses were repeated at stage 13, when axons are starting to project, and the results were essentially the same (Supplementary Fig. 8). Together, our data indicated that the expression of Fas 2 was reduced in the absence of Puc. This reduction was systemic and could not be assigned to individual domains within the VNC.

Although we cannot fully discard the reduction of Fas 2 expression being in part due to widespread apoptosis observed in *puc* mutants (Supplementary Fig. 4), this does not seem to be the case. A decrease in Fas 2 expression at stage 16 was observed after hyper-activating JNK (Hep$^{CA}$ overexpression) in just the RN2 positive neurons. Although some RN2 cells mis-positioned in this condition, they did not die, and no ectopic apoptosis was observed (Supplementary Fig. 9). Indeed, the number of Dcp-1 positive cells per segment was significantly reduced (**$p$ < 0.001) from 45 ± 2 (*n* = 6) to 32 ± 5 (*n* = 10). Thus, cell death and Fas 2 downregulation are disentangled.

We then evaluated if the reduction of Fas 2 expression upon JNK signaling hyperactivation was cell-autonomous. We found a 25 % reduction in Fas 2 expression at the targeted RN2 cells, with only marginal changes outside the *RN2-Gal4* expressing area (Supplementary Fig. 10). Further, the effect of the JNK pathway on Fas 2 expression was also segmentally autonomous. Employing a parasegment specific *PS8/13-Gal80* transgene[33], we locally silenced the overexpression of Hep$^{CA}$ (driven by *RN2-Gal4*) in segments A4–A9 resulting in embryos showing normal Fas 2 levels in posterior segments but reduced expression anteriorly (Supplementary Fig. 11). Altogether, these data unambiguously showed that the regulation of Fas 2 by the JNK pathway was direct and autonomous.

## Fas 2, downstream of JNK signaling, is involved in VNC structural refinement

To explore the potential roles of Fas 2 on the architectural organization of the VNC and its condensation, we employed a series of genetic combinations between hypomorphic (*Fas 2$^{E76}$* and *Fas 2-GFP$^{CB03613}$*) and null (*Fas 2$^{EB112}$*) alleles. We found consistent, although mild, VNC condensation defects that associated to an extensive defasciculation of Fas 2 positive longitudinal tracts. While structural 3D nodes were still present, the 3D correlation along the AP axis was altered (Fig. 4f, g).

Condensation was reduced and VNC length increased (Fig. 4h). Interfering in Fas 2 expression in just the RN2 neurons led to equivalent structural and condensation defects (Supplementary Fig. 12). Hence, Fas 2 stands as an important contributor to the structural refinement of the VNC. Significantly, in *Fas 2$^{E76}$* embryos, phospho-JNK labeling dropped in the VNC (Supplementary Fig. 13), indicating the existence of compensatory mechanisms mediated by the JNK signaling cascade sustaining adequate Fas 2 levels during VNC morphogenesis (see Discussion).

Lastly, we attempted to rescue *puc$^{E69}$* mutant embryos by over-expressing the full-length Fas 2$^{PEST+}$ isoform with the pan-neural driver *Elav-Gal4*[34]. The *Elav-Gal4* is active in all post-mitotic neurons, including the Fas 2-expressing early born pioneers aCC, pCC and RP2 and VUM neurons. Control animals overexpressing Fas 2 in a wild type background showed no effect on VNC structural organization and/or condensation. Yet, we found that in the *puc$^{E69}$* homozygous mutant background, the pan-neural overexpression of Fas 2 failed to suppress the VNC structural condensation defects (Supplementary Fig. 13). This indicates that additional JNK-regulated elements contributing to the correct structural organization of the VNC must exist (see Discussion).

Summarizing, Fas 2 expression in a subset of pioneer neurons (aCC, pCC and RP2) is essential for proper VNC architecture and condensation. Yet, Fas 2 is not the unique mediator of the JNK pathway in its role in VNC architecture refinement.

## Cell-cell adhesion in VNC structural organization

To explore for alternative mediators of the JNK role on architecturally shaping the VNC, we turned to Neurexin IV (Nrx IV), a septate junction element, expressed in glia and neurons. Septate junction components are regulated by JNK signaling in different morphogenetic processes[35] and it is known that Nrx IV has an early function ensuring the integrity of the neural lamella and the Blood Brain Barrier (BBB)[36], which is an essential requisite for VNC condensation[37]. In addition, Nrx IV has a major role on the ensheathment of some axons by glial cells, ensuring the separation of single axons or axonal fascicles[38]. A major consequence of the loss of Nrx IV is the over compaction of segmental commissures as a consequence of a failure in glial cell migration[39]. We examined the expression of Nrx IV upon interfering in JNK signaling and found that by stage 16, Nrx IV was present at low levels in most neurons in the neuropile (Fig. 5a). Nrx IV expression in *puc* mutant embryos (*puc$^{E69}$*), opposite to Fas 2, significantly increased (Fig. 5b, c) and an equivalent response was observed after altering the level of JNK activity in RN2 neurons (Fig. 5c–f). The autonomous inhibition of the pathway by the overexpression of Bsk$^{DN}$ led to a reduction of Nrx IV levels, while its hyperactivation, upon Hep$^{CA}$ overexpression, increased its expression. Interestingly, at stage 13, this regulatory link was reversed and Nrx IV expression decreased in response to JNK signaling activation (Supplementary Fig. 8) (see Discussion). In summary, the regulated activity of the JNK pathway in longitudinal connectives and motoneuron ISN pioneers is essential for the control of Fas 2 and Nrx IV expression dynamics and the correct conformation of the VNC cell adhesion landscape.

## JNK signaling regulation of cell adhesion components is mediated by *zfh1*

To determine how Fas 2 and Nrx IV expression respond to JNK signaling, we searched for potential intermediate regulators. It has been recently shown that in *zfh1* mutants Fas 2 is downregulated in the VNC, and more specifically in the aCC and RP2 ISN pioneers[40]. Further, Zfh1 is sufficient to promote transcription of Fas 2 in some interneurons[41]. Zfh1 is a Zn finger homeodomain family member expressed in all motor neurons, partially overlapping with *puc*. We hypothesized that JNK signaling could affect Zfh1 expression, which, in turn, may modulate Fas 2 and Nrx IV expression levels, finally contributing to the 3D structural organization of the VNC.

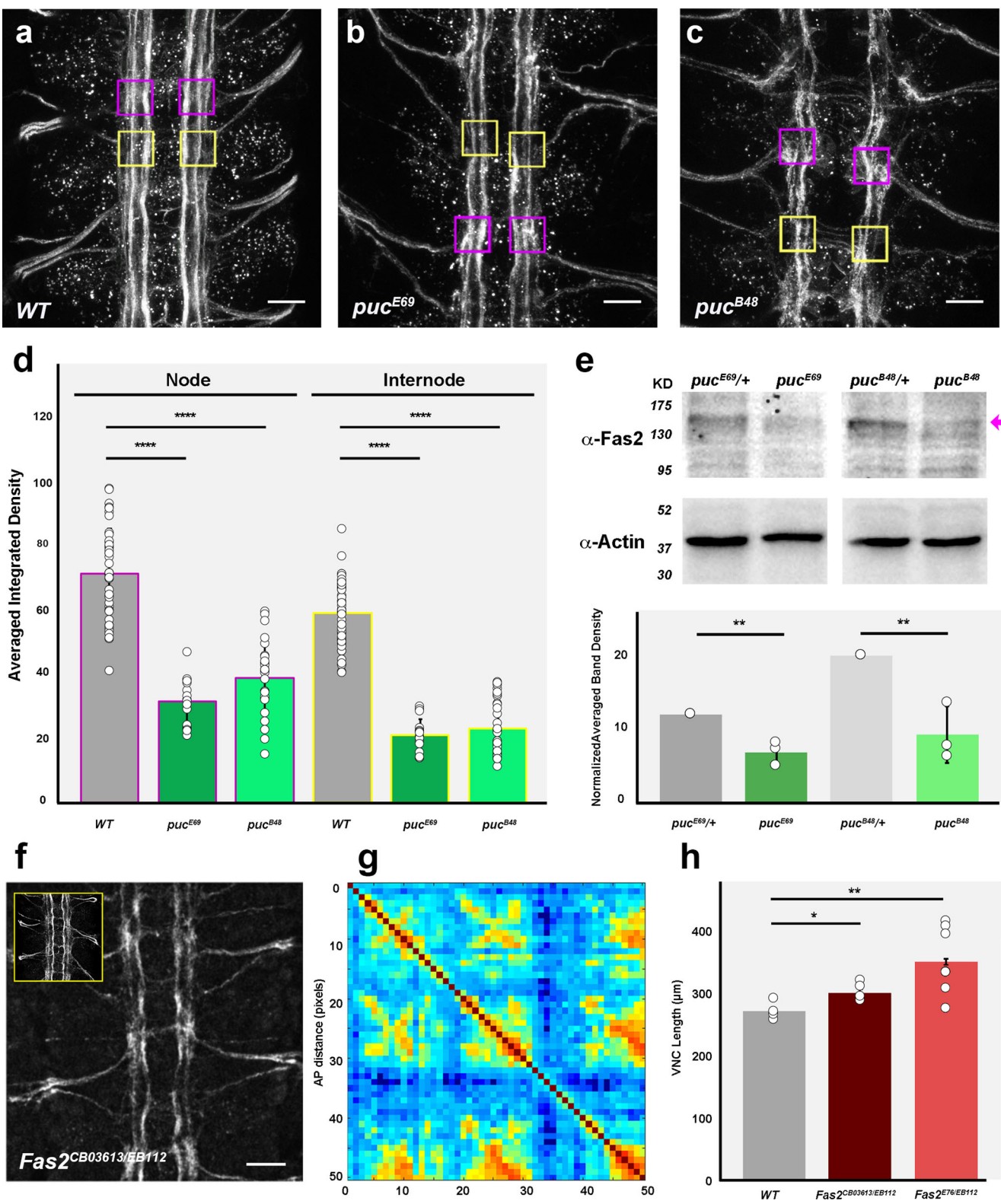

In *zfh1* mutants, we found that the architectural robustness of the Fas 2 connectives was strongly affected and that the VNC displayed an underdeveloped character. Spatial correlations became noisier and internodal distance increased (Fig. 6a–c), recapitulating *puc* (or Hep^CA overexpression in pioneer neurons) phenotypes. Additionally, we observed that the expression of Fas 2 (averaged integrated density in 3D nodes and internodes) was consistently low in *zfh1* (Fig. 6d–f), as it is in *puc* mutants. In short, the failed architectural organization of the

VNC in *zfh1* may be partly understood as a result of Fas 2 down-regulation. We next explored the relationship between *puc* and *zfh1*. Zfh1 expression in *puc* mutants showed a systemic reduction, being shut down in subsets of neurons (Fig. 6g–i). Overall Zfh1 levels also decreased after the hyperactivation of the JNK pathway by Hep^CA in RN2 cells, while they increased after Bsk^DN overexpression (Supplementary Fig. 14). Remarkably, in *zfh1* embryos the activity of the JNK pathway was also significantly increased (phospho-JNK levels)

**Fig. 4 | Fas 2 is necessary for VNC condensation and its expression is modulated by the JNK pathway. a** wild type, **b** $puc^{E69}$ and **c** $puc^{B48}$, stage 16 embryos, Fas 2 immunoreactivity. Maximum projection of ventral views across three abdominal segments. Overlaid squares represent the ROIs used for measuring the Average Integrated Density at nodes (magenta) and internodes (yellow). Anterior is up. Scale bar 10 μm. **d** Quantification of the Average Integrated Density of Fas 2 signal at nodes (magenta perimeter) and internodes (yellow perimeter). Wild type (grey) (Nodes, $n = 43$; Internodes, $n = 49$); $puc^{E69}$ (dark green) (Nodes, $n = 16$; Internodes, $n = 16$); $puc^{B48}$ (light green) (Nodes, $n = 26$; Internodes, $n = 32$). Data presented as mean ± SD. Parametric Student $t$-tests were employed. Significant differences ($^{****}p < 0.0001$) were detected in all cases. **e** Immunodetection of Fas 2 in extracts from stage 16 embryos. The Fas 2 antibody detected a band at 140 kD (magenta arrow), the intensity of which was reduced in $puc$ embryos. Actin (>42 kD) served as loading control. Averaged Fas 2 band intensities were normalized to actin levels and to interblot signal variations. The wild type ($puc^{E69}$ / + or $puc^{B48}$ / +) signal level was employed as a reference. $puc^{E69}$ / + ($n = 3$); $puc^{E69}$ ($n = 3$); $puc^{B48}$ / + ($n = 3$); $puc^{B48}$ ($n = 3$). Data presented as mean ± SD. Parametric Student $t$-tests were employed. Significant differences in Fas 2 levels ($^{*}p = 0.0011$ for $puc^{E69}$ and $^{**}p = 0015$ for $puc^{B48}$) were detected. **f** Fas 2-GFP pattern of a stage 16 Fas 2 mutant embryo ($Fas\ 2^{CB03613}$ / $Fas\ 2^{EB112}$). Maximum projection of ventral views across three abdominal segments. Axonal network is disrupted [compare to the Fas 2-GFP expression in heterozygous ($Fas\ 2^{CB03613}$ / + ) (inset)]. Scale bar 10 μm. **g** Self-cross-correlation matrix along the AP axis of the Z sections of the image in (**f**). Scales and colormap as in Fig. 1. One disperse node per segment was detected. **h** Quantification of the VNC length in μm. Data presented as mean ± SD. Parametric Student $t$-tests were employed. Significant differences in length ($^{*}p = 0.0172$ and $^{**}p = 0.0011$) were detected between wild type ($n = 4$) and $Fas\ 2$ allelic combinations ($Fas\ 2^{CB03613}$ / $Fas\ 2^{EB112}$ (dark red) ($n = 5$); $Fas\ 2^{E76}$ / $Fas\ 2^{EB112}$ (light red) ($n = 8$)). Source data provided as a Source Data file.

(Fig. 6j–l), revealing a negative regulatory loop mediated by Zfh1 tuning the levels of JNK activity.

Our analyses support a model in which the architectural organization and condensation of the VNC are partly the result of the deployment of Fas 2 in longitudinal connectives and motoneuron ISN pioneers (at least aCC, pCC and RP2) in response to Zfh1, under the control of the JNK pathway. Surprisingly, Nrx IV expression decreased in $zfh1$ embryos as Fas 2 expression (Fig. 5g–j), contrary to its observed upregulation in $puc$ mutants. This point to a complex cross-regulation between the JNK pathway and Zfh1 transcriptional activity (see Discussion). VNC length is affected, not just by hyperactivating JNK signaling or reducing JNK expression (Fig. 5k), but also by inhibiting the expresion of $zfh1$ (Fig. 5l), pointing to a mesoscopic morphogenetic role for this regulatory network.

## Discussion

### The embryonic VNC has a robust architecture
During VNC morphogenesis, both neurons and glial cells grow, change shape, and rearrange over time to generate a mechanically balanced structure. Functionally, the neuropile is stratified, with dendrites of motor and sensory neurons terminating and branching in characteristic domains. Yet, anatomically, it shows few signs of organization, apart from certain regularities such as the commissures and a set of longitudinal axon bundles. This paradox may originate in a shortage of knowledge about the structural cues facilitating networking. Using correlative spatial analyses, we have identified a series of anchoring positions (3D nodes) with a potential mechanical role. The 3D nodes consist of sets of Fas 2 positive axons organized in a stereotyped 3D conformation that is repeated segment by segment (Fig. 1 and Supplementary Fig. 1). We speculate that in architectural terms the longitudinal and commissural connectives serve as structural beams and that 3D nodes may act as cross-points that help to evenly distribute any load from compression, tension or shear that could arise as the VNC forms (or that it may suffer once functional[42]). This robust organization may respond to evolutionary conserved needs in terms of mechanics and physical balance[1,43]. The Fas 2 positive 3D nodes most probably represent only a subset of the elements sustaining the tensional network of the VNC, since other Fas 2 negative axons may organize in equivalent structures.

### The JNK pathway in pioneer neurons modulates VNC architectural organization and condensation
Several studies have shown the important function of JNK signaling in axonal plasticity, stability or regeneration[44,45]. However, its roles in directing CNS morphogenesis and functional coordination have not been addressed.

We found that autonomously increasing or decreasing JNK activity in early-specified midline VUMs and aCC, pCC and RP2 pioneers affects the axonal scaffold as a whole (Fig. 3). These neurons (Fig. 2) are specified and singled out at stage 9 and initiate axonogenesis around stage 11[46]. As the CNS develops, each neuron makes a number of stereotypic pathfinding decisions, navigating to their particular synaptic targets. Axonal scaffold formation initiates when the early ipsilateral interneurons MP1, dMP2, vMP2 and pCC pioneer the longitudinal fascicles (stage 12–13). These primary axonal pathways are subsequently joined by other ipsilaterally and contralaterally projecting axons, with anterior or posterior directionalities. Orchestrated axonal fasciculation / de-fasciculation events (stage 14 and 15) along with glial cells intercalation between axonal tracts, result in the distribution of longitudinal axons in three sets (medial, intermediate and lateral) in the neuropile (stage 16). At stage 17, neuronal cell bodies relocate and align linearly along the AP axis[47]. This lengthy and complex process appears to be affected when JNK activity levels are altered at early stages of development[42]. First, the proper allocation of pioneer neurons along the neuropile and the development of their axonal paths become abnormal. Then, their inaccurate early axonogenesis apparently misleads the fasciculation of follower neurons' axons that encounter an aberrant guidance landscape. Lastly, as a result, the execution of the VNC architectural plan and its condensation are challenged (Fig. 7b).

How JNK signaling becomes activated in a specific subset of longitudinal connectives and ISN pioneers is not known. $Bsk$ mRNA levels in the CNS are dramatically reduced in $eve$ mutant embryos[40], and $eve$ is expressed in most $puc$ positive neurons (presumably neurons with high JNK activity), in particular in RN2 pioneers. When $eve$ expression is completely eliminated from the RP2, aCC and pCC neurons, these pioneers show aberrant axonal morphologies and positions but the longitudinal fascicles are not affected[48]. Yet, an in depth inspection of these data reveals that when the RN2 neurons are $eve$ null the scaling of the axonal scaffold (the position of the anterior and posterior commissures and the inter-commissural space) are aberrant and VNC condensation is defective. This suggests, in the absence of confirmatory self-cross-correlation analyses, that the distribution or integrity of Fas 2 bundles and the structural organization of the neuropile are affected in a rather similar way, expected considering the negative loop implemented by Puc, to that of the $RN2\text{-}Gal4> UAS\text{-}Hep^{CA}$ embryos. Indeed, it remains to be tested if Eve is upstream of JNK signaling in these neurons.

### Cell-cell adhesion instructs VNC condensation
Both our data (Fig. 4), and previous literature (*e.g.*, the role of DSCAM in the regulation of circuit level plasticity of retinal bipolar cells[49]), point to an important role for cell-cell adhesion components in the structural organization of the CNS. In particular, changes in cell-cell adhesion components (Fas 2 and, possibly, Nrx IV) in a specific subset of neurons (*e.g.*, those targeted by $RN2\text{-}Gal4$ and $MzVum\text{-}Gal4$) contribute, at a mesoscopic level, to the global topographical organization of the VNC. Fas 2 deficit, as a result of an increase in JNK activity, leads

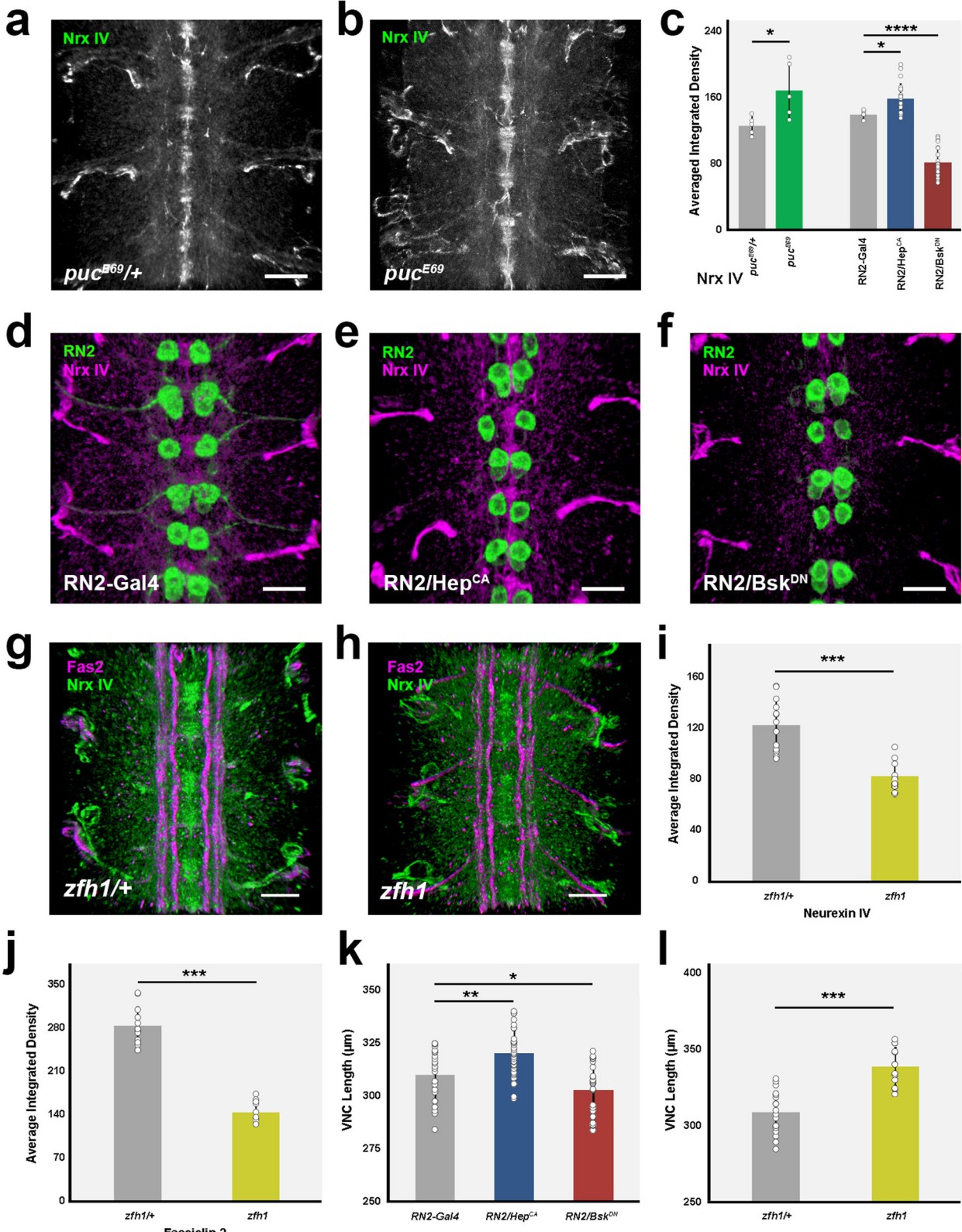

to axonal misrouting, and also to aberrant dendritic arborization[42]. As a consequence, the organization of the local axonal network is disrupted. The shape and compaction of individual segments are affected, which cumulatively perturbs the global architecture and condensation of the VNC (Fig. 7a). Such a direct link between JNK signaling and adhesion elements has been noticed in other related scenarios. For example, in the third instar *Drosophila* larvae, Bsk promotes the axonal pruning of mushroom body gamma neurons by reducing membrane levels of Fas 2[50].

Although our genetic analyses show that Fas 2 is essential for the structural organization and condensation of the VNC, the failure of Fas 2 on rescuing the loss of Puc activity (Fig. 4) points to the contribution of other factors, potentially Nrx IV, downstream of JNK. Nrx IV is present on the membranes of all neurons and at high levels at interfaces

**Fig. 5 | Nrx IV expression is modulated by JNK signaling and Zfh1 activity.**
**a** *puc^E69^*/+ and **b** *puc^E69^*, stage 16 embryos, Nrx IV immunoreactivity. Nrx IV increases in *puc* mutants. Anterior is up. Scale bar 10 μm. **c** Nrx IV Average Integrated Density per segment for (**a**) (grey) (*puc^E69^*/+, *n* = 5); (**b**) (green) (*puc^E69^*, *n* = 5); (**d**) (grey) *RN2-Gal4* /+, *n* = 5; (**e**) (blue) *RN2-Gal4 > UAS-Hep^CA^*, *n* = 21; and (**f**) (red) *RN2-Gal4 > UAS-Bsk^DN^*, *n* = 22. Data presented as mean ± SD. Parametric Student *t*-tests were employed. Significant differences in Nrx IV levels were detected. *p* = 0.0277, *puc^E69^*/+ vs *puc^E69^*; *p* = 0.022, *RN2-Gal4* /+ vs *RN2-Gal4 > UAS-Hep^CA^*; and ***p* < 0.0001, *RN2-Gal4* /+ vs *RN2-Gal4 > UAS-Bsk^DN^*. **d** *RN2-Gal4* /+, **e** *RN2-Gal4 > UAS-Hep^CA^* and **f** *RN2-Gal4 > UAS-Bsk^DN^* stage 16 embryos. Sum projections of ventral views across three abdominal segments (RN2 - green). Autonomous hyperactivation of the pathway led to an increase of Nrx IV levels (magenta), while its inhibition reduced its expression. Anterior is up. Scale bar 10 μm. **g** *zfh1* /+ and **h** *zfh1* stage 16 embryos with midline glia[38]. Neuronally-expressed Nrx IV, together with the

Fas 2 (magenta) and Nrx IV (green) immunoreactivity. Scale bar 10 μm. **i, j** Average Integrated Densities per segment of Nrx IV and Fas 2 for (**g**) (*zfh1* /+) (grey) (Fas 2, *n* = 12; Nrx IV, *n* = 13) and (**h**) (*zfh1*) (dark yellow) (Fas 2, *n* = 10; Nrx IV, *n* = 10). Data presented as mean ± SD. Parametric Student *t*-tests were employed. Significant differences (***p* < 0.001) for Fas 2 and Nrx IV levels were observed. **k** VNC length in μm for *RN2-Gal4* /+ (grey) (*n* = 24), *RN2-Gal4 > UAS-Hep^CA^* (blue) (*n* = 34) and *RN2-Gal4 > UAS-Bsk^DN^* (red) (*n* = 27) stage 16 embryos. Data presented as mean ± SD. Parametric Student *t*-tests were employed. Significant differences in length were detected upon hyperactivation (***p* = 0.0018) or inactivation (*p* = 0.0364) of the pathway. **l** VNC length in μm for *zfh1* /+ (grey) (*n* = 18) and *zfh1* (dark yellow) (*n* = 13) stage 16 embryos. Data presented as mean ± SD. Parametric Student *t*-tests were employed. Significant differences in length were detected (***p* < 0.001). Source data provided as a Source Data file.

midline glia-expressed Ig-domain protein Wrapper, assemble in heterophilic complexes mediating glia-neuron interactions and ensuring the glial wrapping of the CNS[51]. In *Nrx IV* mutants, commissure separation fails from stage 13 onwards[38]. This failure is due to the inability of the midline glia to properly migrate and to send projections into the commissures. As a consequence, commissures physically constrict[39], which could contribute to VNC over-condensation. The increase in Nrx IV levels observed in *puc* mutants, and upon overexpression of Hep^CA^ in RN2 cells, might have an opposite effect, *i.e.*, the over-recruitment of glia towards the midline, sterically impeding condensation. This may be a conserved function as, at paranodal axoglial junctions in mice, the Nrx IV homolog Caspr binds in trans to the glia Neurofascin, promoting myelinization[52]. Consistently, the local downregulation of Fas 2 mediated adhesion between neurons and glia has been shown to be required to initiate glial cell migration in the embryonic peripheral nervous system[53].

## Zfh1 decodes JNK signaling activity into cell behavior modulation

How the activity of the JNK signaling pathway in just a few longitudinal connectives and ISN pioneers results in global mesoscopic defects during VNC condensation is unclear. We provide evidence that the JNK activity alters neurons adhesive properties modulating Fas 2 and Nrx IV expression, via Zfh1. This, itself, implements a negative regulatory loop on JNK signaling providing a mean for the dynamic control of its activity (Fig. 7a). An equivalent cross-interaction has been shown for the regulation of cell survival in peripheral glia[54].

Epistasis and rescue analysis could further elucidate the molecular links between the JNK cascade with the Zfh1 transcription factor and with the cell adhesion elements implicated, those examined here or other yet unidentified. However, considering that the establishment of the VNC 3D architectural organization does not rely only on the combinatorial modulation of adhesion and guidance elements by the JNK pathway, but also on their coordinated activities and on multiple regulatory transcriptional loops, epistatic assays would hardly provide any major hierarchical clue. Fas 2 just partly rescued *puc* loss of function and, most probably, any other single rescue attempt would give the same result.

Beyond adhesion, JNK activity may also affect neurons shape and/or migration[55]. While, for technical reasons, cells' shapes cannot be determined within the dense neuropile, we observed that cell bodies intercalation during compaction was highly stereotyped[9]. Migrating neuron precursors, which are formed in specific locations within the developing CNS, travel short distances to occupy distinct positions within the mature brain or nerve cord. This late positioning has a deterministic role in its connectivity and, indeed, perturbations in neuronal migration are known to cause neurodevelopmental defects[56]. In *zfh1* mutants, the migratory path of the JNK and *zfh1* positive, RP2 ISN pioneer (derived from the NB4-2 lineage) is affected. Interestingly, *zfh1* also affects, non-autonomously, the migration of the RP2 sibling neuron, which does not express *zfh1*[55], and the axonal projections of

motoneurons and interneurons of other neuroblast lineages, such as the NB7-3[57]. *zfh1* thus exhibit a non-autonomous instrumental role on the architectural organization (cell bodies allocation and axonal projections) of the immature nerve cord. This scenario is consistent with the phenotypic defects observed upon interfering with JNK signaling in RP2 and other pioneers. Indeed, we also identified upon JNK hyperactivation both consistent and reproducible mis-localization of the RP2 (see Fig. 3) and non-autonomous axonal misrouting defects.

## VNC condensation is a complex process subjected to multifactorial regulation

Mechanically, VNC morphogenesis relies on a segmentally iterated axonal scaffold whose geometrical integrity is essential for the active oscillatory constrictions leading to condensation[9]. This scaffold is built following the paths of pioneer neurons[46]. We found that JNK signaling is instrumental for its structural organization by modulating the adhesion properties of a subset of longitudinal connectives and ISN pioneers. Remarkably, killing these pioneers does not deeply affect Fas 2 longitudinals and major disruptions can only be observed upon elimination of all pioneer neurons[23]. This implies that there are qualitative differences between pioneers' death, and the alteration of their interpretative and integrative capacities in a developing environment. By interfering in the JNK cascade, we are generating phenotypes that stand apart from the classical role of pioneers in axon guidance, and that are related to something different, the 3D structural organization of the neuropile.

Other elements, such as glia or haemocytes, also contribute to VNC condensation[37]. While *puc* is expressed in VNC neurons and haemocytes, but not in glia, our interference experiments with the neuron specific MzVum and RN2 drivers, convincingly discard any direct role of JNK signaling in haemocytes influencing the axonal scaffold architecture. Haemocytes and glia may still be mechanically important, and we have shown that interfering in glia contractility abolishes VNC condensation[9]. A potentially important role for haemocytes could be to participate in ECM deployment and turnover. The ECM surrounds the CNS and may act as a frictional substrate for tissue rearrangement and/or as a physical barrier restraining tissue shape[9]. Its real role remains to be determined.

Finally, it is important to highlight the significance of the role of the pCC neuron. This is the single longitudinal connective pioneer affected when interfering in JNK signaling employing the *RN2-Gal4* line. Our data strongly support an early instructive "structural" role for the pCC, probably in coordination with other elements. Among these, we cannot exclude, *prima facie*, the contribution of the aCC and RP2 ISN pioneers. They may indirectly affect the organization of the neuromere around the 3D structural nodes. The aCC and pCC cell bodies are located in the proximity of the nodes and positional and morphological alterations on the RP2 motoneuron pioneer can affect adjacent interneurons. In this scenario, the structural organization and condensation defects observed may not be caused just by a single longitudinal connective pioneer, the pCC, but by a seeding effect of the rest of the RN2 early born neurons. Yet, we are quite certain that other

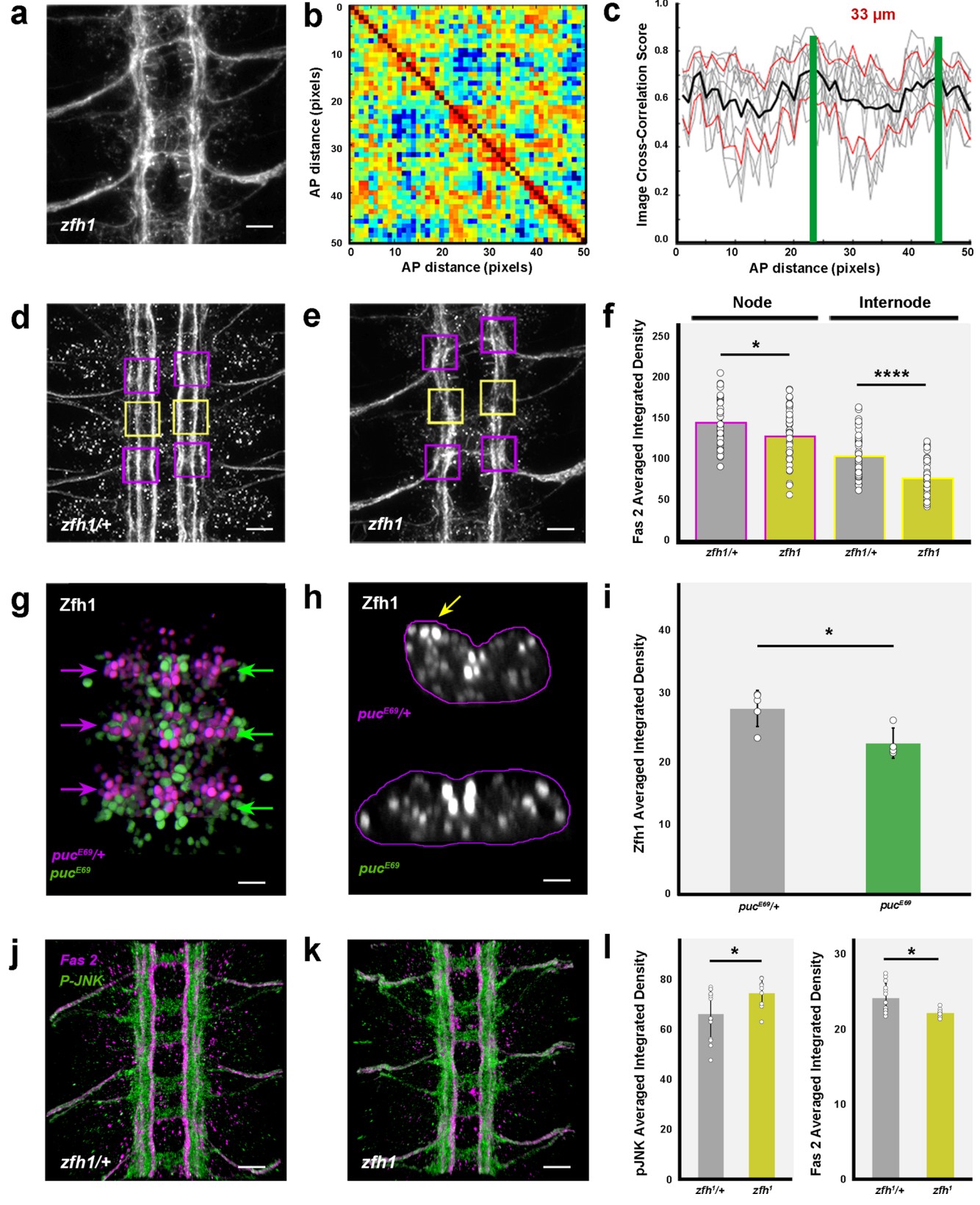

neurons (maybe of basal JNK activity), not studied in this work could also have an important role on VNC morphogenesis.

## Methods
### Drosophila strains
The following stocks were used:

w1118; pucE69LacZ / TM3, twi-GFP[19]
w1118; pucB48LacZ / TM3, twi-GFP[19]
w1118; UAS-GFP; pucE69I–Gal4 / TM6B[20]
w1118; UAS-GFP; pucB48–Gal4 / TM6B[20]
w1118; P{w[+mC]=UAS-bsk.DN}2 (BDSC #6409)
w1118; P{w[+mC]=UAS-Hep.Act}2 (BDSC #9306)

**Fig. 6 | JNK signaling modulates the expression of Zfh1, which feedbacks on JNK activity and affects Fas 2 and Nrx IV expression. a** Maximum projection of the VNC of a stage 16 *zfh1* embryo stained with Fas 2. Anterior is up. Scale bar 10 μm. **b** Self-cross-correlation matrix of the Z sections of (**a**) along the AP axis. Scales and colormap as in Fig. 1. Nodes are sustained but structural noise increases dramatically. **c** Image cross-correlation score along the AP axis for (**b**) (*n* = 8 embryos). 3D nodes fall 33 μm apart and the internodal correlation flattens. The grey lines show individual profiles, black averaged values and red ± SD. Green bars indicate maximum correlations (3D nodes). **d** *zfh1* / + and e) *zfh1* stage 16 embryos Fas 2 immunoreactivity. Maximum projection of ventral views across three abdominal segments. Overlaid squares [nodes (magenta) and internodes (yellow)] are the ROIs employed on Average Integrated Density measurements. Scale bar 10 μm. **f** Average Integrated Density at nodes and internodes (magenta and yellow perimeters) for (**d**) (grey) (Nodes, *n* = 41; Internodes, *n* = 44) and (**e**) (dark yellow) (Nodes, *n* = 43; Internodes, *n* = 42)). Data presented as mean ± SD. Parametric Student *t*-tests were employed. Significant differences in Fas 2 ($^*p$ = 0.0174 and

$^{****}p$ < 0.0001) were observed at nodes and internodes between *zfh1* / + and *zfh1*. **g** Overlay of aligned 3D reconstructions of the VNC of *puc*$^{E69}$ / + (magenta) and *puc*$^{E69}$ (green) stage 16 embryos immunostained for Zfh1. Arrowheads point to segmental midpoints. Scale bar 10 μm. **h** Z cross-sections from (**g**) at segmental midpoints. The VNC of *puc*$^{E69}$ shows dramatic flattening. Several neurons lose Zfh1 (yellow arrow). Scale bar 10 μm. **i** Zfh1 Average Integrated Density per segment on *puc*$^{E69}$ / + (grey) (*n* = 5) and *puc*$^{E69}$ (dark green) (*n* = 4) embryos. Data presented as mean ± SD. Parametric Student *t*-tests were employed. A significant decrease ($^*p$ = 0.0175) was detected for *puc*$^{E69}$. **j** *zfh1* / + and k) *zfh1* stage 16 embryos Fas 2 (magenta) and P-JNK (green) immunoreactivity. Anterior is up. Scale bar 10 μm. **l** Average Integrated Density per segment for (**j**) (Fas 2 (*n* = 13); P-JNK (*n* = 14)) and (**k**) (Fas 2 (*n* = 7); P-JNK (*n* = 9)). Data presented as mean ± SD. Parametric Student *t*-tests were employed. P-JNK levels increase ($^*p$ = 0.0356) and Fas 2 decrease ($^*p$ = 0.0214) from *zfh1* / + (grey) to *zfh1* (dark yellow). Source data provided as a Source Data file.

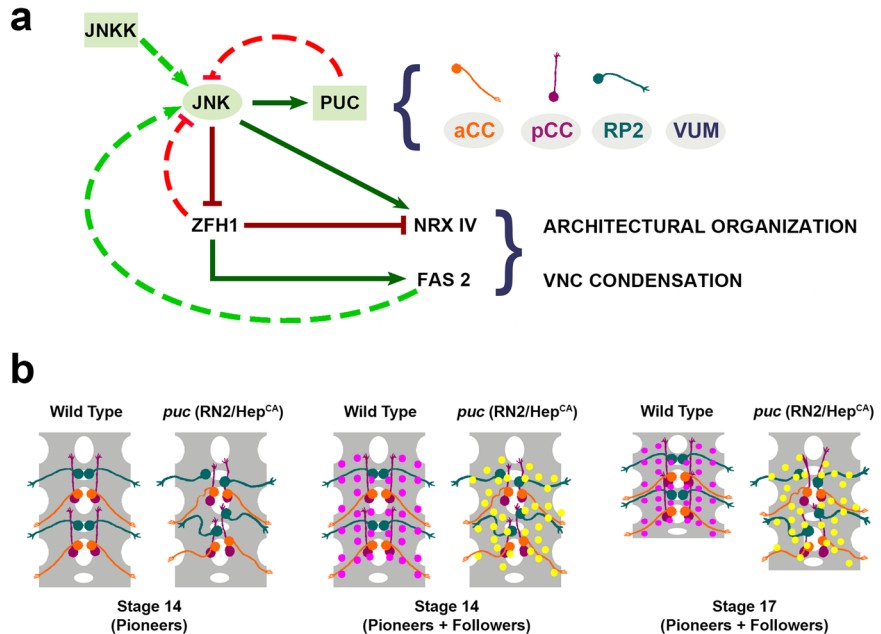

**Fig. 7 | JNK signaling activity in pioneer neurons modulates the architectural organization of the CNS. a** Regulatory network modulating VNC architectural organization and condensation. Precise JNK activity levels in early-specified neurons (aCC, pCC, RP2 and VUMs) are regulated by a primary negative feedback loop mediated by Puc. Excessive JNK activity in *puc* mutants or upon early overexpression of Hep$^{CA}$ (constitutively active JNKK) in aCC, pCC, RP2 and/or VUMs neurons leads to a general downregulation of Fas 2 (mainly autonomous) and an increase in Nrx IV expression. This affects the general architectural robustness of the VNC, preventing its condensation. Zfh1 acts as an intermediate factor in pioneers regulating Fas 2 and Nrx IV expression in response to JNK activity. On the other hand, Zfh1, negatively, and Fas 2, positively, affect JNK activity, establishing secondary control loops. Regulatory links at the level of expression control are indicated by continuous lines (Dark Green - positive; Dark Red- negative). Links at the level of activity control are represented by discontinuous lines (Light Green -

positive; Light Red - negative). Schematic diagrams of the aCC, pCC, RP2 and VUM neurons are color coded as in (**b**). **b** The modification of the canonical topography and axonal paths of pioneer neurons by alterations in JNK activity levels disturbs their instructive structural roles. The RN2 positive longitudinal connectives and motoneuron ISN pioneers (aCC, pCC and RP2 - color coded as in (**a**) locate in precise positions of the neuropile and project axons with stereotyped trajectories. In *puc* mutants or after overexpressing Hep$^{CA}$, their positions are altered (mostly the RP2) and the paths of their axons distorted[42] (Stage 14 Pioneers). This results in the repositioning of followers (magenta in the wild type and yellow in mutant conditions), adapting to the aberrant axonal landscape (Stage 14 Pioneers and Followers). One secondary effect of the disorganization of the architecture of the VNC, along other defects in neuron/glia interactions, is its failure to condense (Stage 17).

---

*P{GawB}elav[C155]* (BDSC #6923)
*w1118; P{w[+mC]=eve-Gal4.RN2}T, P{w[+mC]=UAS-mCD8::GFP.L}; P{w[+mC]=eve-Gal4.RN2}G, P{w[+mC]=UAS-mCD8::GFP.L}* (Dr. Irene Miguel-Aliaga)
*w1118; P{w[+mC]=eve-Gal4.RN2}O* (Dr. Richard Baines)
*y1 w1118; P{w[+mC]=CQ2-Gal4}O, P{w[+mC]=UAS-mCD8::GFP.L}LL6* (BDSC #7465)
*w1118 P{w[+mW.hs]=GawB}MzVUM; P{y[+t7.7]w[+mC] = 10xUAS-IVS-mCD8::GFP} attP40* (Dr. Irene Miguel-Aliaga)
*w1118; P{UAS-syt.eGFP}2* (BDSC #6925)
*w1118; UAS-Fas 2$^{PEST+}$ / CyO [Nuc::GFP]*[34] (Dr. Brian McCabe)

*w1118, Fas 2$^{E76}$* (Dr. Brian McCabe)
*w1118, Fas 2$^{EB112}$ / FM7[Actin::GFP]* (Dr. Brian McCabe)
*w1118, Fas 2-GFP$^{CB03616}$* (Dr. Christian Klämbt)
*w1118; UAS-GFP; HCJ192(iab3)-Gal80 (PS8/13-Gal80)*[33] (Dr. Wellcome Bender)
*yw; Sp / CyO; 20xUAS-degrad-GFP / TM6B* (Dr. Brian McCabe)
*w1118, Fas 2-GFP$^{CB03616}$ / FM7;; 20xUAS-degrad-GFP / TM6B* (this work)

In all cases, unless otherwise stated, embryos of the *w1118* strain served as controls.

## Genetics

All crosses were performed at room temperature and after 48 h were shifted to different temperatures as the individual experiments required. After the temperature shifts the embryos were processed for immunocytochemistry or for live imaging.

Local inhibition of Hep[CA] overexpression in parasegments 8 to 13 in RN2 neurons was achieved employing a *w1118; UAS-GFP; HCJ192(iab3)-Gal80 (PS8/13-Gal80)* transgenic line expressing the Gal4 inhibitor Gal80 under the control of the *iab3* promoter. The expression of transgenes (*UAS-GFP ± UAS-Hep[CA]*) in the presence of *RN2-Gal4* (*RN2[T]; RN2[G]*) was limited to the anterior parasegments 1 to 7.

To generate local inhibition of Fas 2 expression in RN2 neurons we crossed virgins *Fas 2-GFP[CB03616] / FM7;; UAS-degrad-GFP / TM6B* with males *RN2[O]-Gal4*. Fas 2 in embryos of the genotype *Fas 2-GFP[CB03616]; RN2[O]-Gal4> UAS-degrad-GFP* was specifically degraded in RN2 positive cells.

## Immunohistochemistry

Immunostaining of flat-prepped stage 16 *Drosophila* embryos was performed using the following primary antibodies: mouse anti-Fas 2 (1:100, clone 1D4, DHSB), mouse anti-axons CNS (1:100, BP 102, DHSB), mouse anti-Futsch (1:1000, 22c10, DHSB), rabbit anti-phospho-JNK (Thr[183] / Tyr[185]) (1:100, Cell Signaling #9251), rabbit anti-Dcp-1 (Asp216) (1:100, Cell Signaling #9578), rabbit anti-GFP tag polyclonal (1:600, Thermo Fisher Scientific), rat anti-Elav (1:1000, clone 7E8A10, DHSB), mouse anti-Repo (1:100, clone 8D12 DHSB), mouse anti-Even-skipped (1:100, clone 3C10, DHSB), mouse anti-Engrailed (1:100, clone 4D9, DHSB) and TRITC-conjugated goat anti-HRP (1:200, Jackson ImmunoResearch #123-025-021).

The secondary antibodies used for detection were: Goat anti-Rabbit IgG (H + L), Alexa Fluor 488 conjugate (A-11008), Goat anti-Rabbit IgG (H + L) Alexa Fluor 555 conjugate (A-21428), Goat anti-Mouse IgG (H + L) Alexa Fluor 488 conjugate (A-11001), Goat anti-Mouse IgG (H + L) Alexa Fluor 555 conjugate (A-21422) and Goat anti-Rat IgG (H + L) Alexa Fluor 555 conjugate (A-21434). All secondary antibodies were used in a dilution of 1:600 and were from Invitrogen.

## Sample preparations for immunodetection and image acquisition

*Drosophila* embryo dissections for generating flat preparations were performed according to ref. [58]. Briefly, flies maintained in apple juice-agar plates at 25 °C were synchronized by repetitive changes of the juice-agar plate, with a time interval of 1.5 h. All embryos laid within this time window were aged for approximately 8 h at 29 C or 22 h at 18 °C, until reaching mid-stage 16 (3-part gut stage). At this point embryos were dechorionated with bleach for 1 min, poured into a mesh and rinsed extensively with water. For dissection, embryos were transferred with forceps on the surface of a small piece of double-sided tape, adhered on one of the sides of a poly-L-Lysine coated coverslip. After orienting the embryos dorsal side up and posterior end towards the center of the coverslip, the coverslip was flooded with saline (0.075 M Phosphate Buffer, pH 7.2). Using a pulled glass needle the embryos were manually devitellinized and dragged to the center of the coverslip, where they were attached to the coated glass with their ventral side down. An incision on the dorsal side of the embryo was performed using the glass needle from the anterior to the posterior end of the embryo. The gut was removed by mouth suction and a blowing stream of saline was used to flatten their lateral epidermis. Tissue fixation was done with 3.7% formaldehyde in saline for 10 min at room temperature. After this point standard immunostaining procedures were followed.

In the single case (Supplementary Fig. 4) where whole mount embryos were immunostained, embryo fixation and devitellinization were done according to ref. [59]. Both whole mount and flat-prepped immunostained embryos were mounted in Vectashield anti-fading medium (Vector Laboratories, USA).

Image acquisition was performed on a Zeiss LSM 700 inverted confocal microscope, using a 40 X oil objective lens (NA 1.3). Z-stacks spanning the whole VNC thickness were acquired applying optimal, with respect to pinhole diameter, sectioning with a step size that varied between 0.2–1 μm. Image processing was performed with Fiji[60].

## Live imaging

Dechorionated stage 13 embryos were glued ventral side down on a MatTek glass bottom dish and they were covered with S100 Halocarbon Oil (Merck) to avoid desiccation. Image acquisition was performed on a Zeiss LSM 700 inverted confocal microscope, using a 25 X oil immersion lens (N.A 0.8, Imm Korr DIC M27). Z-stacks spanning the whole VNC thickness, with a 2 μm step size, were acquired every 6 min for a total of 8 h. Processing of time-lapse data was done with Fiji[60].

## Image analysis

Image analysis and quantification of fluorescence intensity were performed using Fiji[60]. In immunostainings where quantification of the fluorescent intensity was required, extra care was taken so that the same antibody batch was used for staining embryos of different genotypes, while identical confocal acquisition settings were applied.

For calculating the average integrated density of Fas 2 signal in an image, regions of interest (ROIs) of fixed size were measured at the 3D nodes and internodes positions. Average Integrated Density values obtained by measuring ROIs in several animals (n stated in Figure legends) were then pooled for calculating the mean value as well as standard deviations. For those cases where the quantification of integrated density was performed on a segmental basis, designated ROIs covered the whole segment area. To calculate the percentage of autonomous and non-autonomous reduction in Fas 2 signal, the GFP signal (RN2 expression) was thresholded and 3D binary masks were created. Masks were employed to delimit the quantification of Fas 2 levels in different domains.

## Statistical analysis

All statistical analyses were performed using GraphPad Software (GraphPad Software Inc., La Jolla, CA, USA). In all cases one sample *t* test was performed and probability values $p < 0.05$ were considered as significant. In all figures probability values $p < 0.05$ are denoted by *, $p < 0.01$ are denoted by **, $p < 0.001$ are denoted by *** and $p < 0.0001$ are denoted by ****.

## Correlations data analysis

Images were scaled to be isotropic in all axes. Deconvolution was found to not significantly improve the image correlation. Viewed along the AP axis, images were cropped to include only the VNC, and then the VNC was split into 50 bins, corresponding to 1.65 μm length each. Within each bin, a maximum intensity projection was performed along the included (typically eleven) planes in the z-axis. The Matlab function *imregister* was then used to perform the image registration for both inter- and intra- embryo correlations. The optimizer for *imregister* was defined with multimodal metric, tolerance of $10^{-6}$ and 500 iterations. Before image registration, the center of mass of the two images was aligned so to maximize image overlap. The code was run on a PC with 128 Gb memory and in the most demanding case (full embryo correlation: $50 \times 50 = 2500$ correlations) took around one hour per sample.

To create the image correlation traces, the regions of high correlation (*e.g.*, as identified by arrows in Fig. 1) were manually identified and then plotted considering the mean correlation along ± 1 row in the correlation matrices. The diagonal values were ignored. Traces were offset along AP axis to ensure maximum overlap of peaks in the region 25–57 μm to aid comparison between different datasets. In all figures, individual profiles are plotted in gray, with the mean (black) and ±1 standard deviation (red) shown.

To analyze the steepness of the spatial correlations along the AP axis, we created in silico data from the real experimental data points - *i.e.*, each in silico data point was drawn from a distribution defined by the mean and standard deviation in the samples. We determined the best linear fit to each in silico data set and repeated 500 times to find the average gradient fit. This was a robust way to find the steepness as it accounts for experimental noise. To compare between datasets, we defined the variable (where gradient corresponds to the spatial derivative of the correlation along the AP axis):

$$r = (\max(gradient) - abs(\min(gradient)))/[(\max(gradient) + abs(\min(gradient)))/2].$$

### Western blotting

Dechorionated stage 16 embryos of the *puc^{E69}LacZ / TM3, twi-GFP* and *puc^{B48}LacZ / TM3, twi-GFP* lines were used for generating whole embryo extracts for western blot analysis. In detail, homozygous embryos were separated from their balanced siblings under a fluorescent dissecting scope based on the *twi-GFP* expression. Approximately 100 embryos of each type were fully homogenized in 100 μL of iced cold RIPA buffer, supplemented with Protease Inhibitors (Protease Inhibitor Cocktail Tablets, Roche Life Sciences). Subsequently, the extracts were centrifuged at 3500 *g* for 15 min at 4 °C and the supernatants were collected. Total protein concentration in the lysates was determined by Bradford assay (Bio-Rad) and 30 μg per sample were analyzed in 10% SDS-PAGE. BlueEasy Prestained Protein Markers (Nippon Genetics Cat. No. MWP06) were employed. Proteins were blotted on Protran nitrocellulose membranes (Whatman). These were cut by half above the 52 KD marker level and the top and bottom sections were probed, respectively, with anti-Fas 2 (1:1000, clone 1D4, DSHB) and Pan-actin (1:1000, Cell Signaling) antibodies. The ECL Plus chemiluminescent substrate (Pierce, Thermo Fisher Scientific) was used for antibody detection. Western Blot signal quantification was performed with Fiji by measuring the average density of each band.

### Data availability

The authors declare that all data supporting the findings of this study are available within the article and its supplementary information files and are available from the corresponding author on request. Source data are provided with this paper.

### Code availability

The correlation analysis was performed in Matlab using standard Matlab functions. Full code is provided on Github: https://github.com/TimSaundersLab[61].

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

## Acknowledgements

We are very grateful to all colleagues that provided us materials, training and guidance throughout this work as well as for commenting on the manuscript. Amongst them we like to specially thank Elisabeth Knust, Christian Klambt, Claude Desplan, Matthias Landgraf, Chris Doe, Andreas Prokop, Natalia Sanchez-Soriano, Alicia Hidalgo, Brian McCabe and Irene Miguel-Aliaga. We also thank our colleagues at the Parc Cientific de Barcelona, the Mechanobiology Institute of Singapore and the BSRC "Alexander Fleming" of Athens for their constant encouragement and support. Martín-Blanco laboratory was supported by funds from Programa Estatal de Fomento de la Investigación Científica y Técnica de Excelencia (BFU2014-57019-P, BFU2017-82876-P and PID2020-116273GB-I00) and from Fundación Ramón Areces. TES was supported by a Singapore National Research Foundation Fellowship (NRF2012NRF-NRFF001-094), core support from the Mechanobiology Institute and start-up funds from University of Warwick.

## Author contributions

K.K., T.E.S, G.P. and E.M-B. designed the experiments and analyzed the data. K.K. and E.M-B. conducted the experiments. K.K. and G.P. performed the biochemical analysis, and K.K. and T.E.S. performed the analytical correlations. E.M-B. wrote the initial draft. K.K., T.E.S. and G.P. revised the final draft.

## Competing interests

The authors declare no competing interests.
