## [Peer Review File · Nature Communications]

JNK signaling in pioneer neurons organizes ventral nerve cord architecture in *Drosophila* embryosREVIEWER COMMENTS

Reviewer #1 (Remarks to the Author):

Since I am not a neurobiology expert, when reviewing this manuscript the editor asked me to look specifically at the genetic aspects of the work; the nature of the different mutations and the utilization of the Gal4/UAS system to manipulate the expression of the JNK pathway.

The experimental approach of the work is simple and straightforward: having described the overall structure of the ventral nerve cord (VNC) in *Drosophila* embryos, the authors describe how alterations of JNK function impinge on the architecture of the VNC.

To manipulate JNK activity the authors make extensive use of mutations at the *puc* (puckered) gene, a JNK target that functions as a negative regulator of the pathway, as previously shown by the senior author of the work. Thus *puc* mutations cause inappropriate elevation of JNK activity. The authors find that in *puc* mutant embryos the condensation of the VNC is abnormal. Moreover, the authors use a *puc-Gal4* line to locate JNK expression in specific groups of neurons of the VNC.

To establish that the alterations found in the VNC of *puc* mutants are due to excessive JNK activity and not to general disturbance of the mutant embryos, the authors use several Gal4 lines to force high JNK expression (using constitutive active form of the Jun kinase Hemipterous) in the subsets of neurons that normally express *puc*. They find alterations in the VNC consistent with those observed in *puc* mutants. Interestingly, suppression of JNK activity (using a dominant negative form of the kinase Basket) in the same neurons cause similar morphological defects in the VNC.

The authors also make use of *puc* mutations to identify downstream factors that may be associated with the VNC phenotypical alterations. The levels of Fasciclin 2 (*Fas 2*), a cell adhesion molecule, appear to be reduced. Moreover, *Fas 2* mutations affect VNC condensation, suggesting that *Fas 2* mediates JNK activity on VNC condensation. They also find other factors like neurexin and *zfh1*, whose expression is abnormal in *puc* mutants and their loss of function causes defects in VNC architecture.

In sum, the authors make effective use of mutations affecting JNK activity and of Gal4/UAS mediated manipulations to describe the requirements for JNK function in the formation of the *Drosophila* Ventral Nerve Cord. Essentially it is descriptive work, supported by some nice figures and movies; some genetic elements have been identified, but clearly the picture is far from complete. How much advance it represents in the current understanding of the *Drosophila* Nervous System I cannot tell

Reviewer #2 (Remarks to the Author):

In this manuscript Karkali et al use spatial 3D correlation analysis to detect conserved domains within *Fas2* expressing longitudinal axon bundles which they define as structural nodes. The distance among these structural nodes is used to determine defects in ventral nerve cord architecture in different genetic backgrounds. The authors analyze *puk* Loss-of-function, a background where the JNK pathway is activated as well as well as a dominant negative *Bsk* or constitutively active *Hep* driven in small subsets

of neurons. They conclude that the JNK pathway is required for the structural organization of the ventral nerve cord. They further analyze Fas2 expression on those backgrounds to conclude that fas2 is downstream the JNK pathway, although not the only player. Finally, they show how Zfh1 mediates the expression of Fas2 and Nr4 IV in opposite ways downstream the JNK pathway.

The proposed model where JNK regulates Zfh1, which in turn downregulates Nr4 IV and upregulates Fas2 generating an autoregulatory loop at the level of JNK to regulate VNC development could be interesting to a wide audience. Unfortunately, the evidence provided and methodological problems make the author's conclusions premature.

Fig1 -3D correlation analysis:

A better description of the methodology and panels in figure 1 is required. There are axes in the graphs without labels. There should be a color-coded scale for the correlation. Do the double headed arrows actually point to the position in panel A?. There seems to be a problem with the scale bar as the whole neuropil seems to be 60 μ m while panel B, the cross-correlation matrix is 50 bin (1.65 μ m/bin 82.5 μ m total), where the image is cropped to the VNC. It is not clear what does this refer to. Are the bins 1.65 μ m or 1.65 μ m²?. In addition the cross-correlation matrixes seem to correspond to hemisegments as the repeated correlation hotspots seem to correspond to Fas2 fascicles on one half of the embryo. This is not specified anywhere.

Fig3 Targeted activation and inhibition of the JNK pathway:

In this experiment the authors inhibit or hyperactivate the JNK pathway in subsets of neurons. Satisfyingly, when the pathway is tampered with on U motor neurons there are no overt defects in longitudinal Fas2 pathways. However, when the JNK pathway is hyperactivated or inhibited with the RN2-Gal4 driver there seem to be major defects in the VNC. This is quite significant. The RN2-Gal4 driver has been extensively characterized and it drives expression in only 3 neurons, the aCC and RP2 motor neurons that pioneer the ISN and pCC. Such an effect on Fas2 longitudinal fascicles of a single neuron are quite significant and would point towards an instructive role of pCC, that has never been described. In other genetic backgrounds (for example eve nulls on the RN2 neurons), where this interneuron mis-projects across the midline, longitudinal fascicles are properly fasciculated. The effect reported in this manuscript (lines 174-178) is quite severe and may point to a non-cell autonomous effect. This major discrepancy should have been at least discussed.

Figs 4, S8,S9 Fas2 regulation by JNK.

The authors quantify Fas2 fluorescent staining in puc null embryos (fig 4) and conclude that there is a reduction. The authors also point out that in puc mutants there is also an increase in apoptosis (line 131). Apoptosis is not quantified in the present work but become relevant here as a reduction in Fas2 signal could be due to increase in apoptosis of Fas2+ neurons. In an attempt to determine Fas2 regulation in a cell specific manner the authors hyperactivate the JNK pathway specifically in RN2 neurons. In these circumstances they observe a significant Fas2 reduction at nodes and internodes at stage 16. This experiment should have been done at stage 12-13 for various reasons. To have a single cell resolution the experiment has to be performed when axons are starting to project and at those stages Fas2 is clearly present. At stage 16, only the pCC axon (within a bundle of several tenths) has the

JNK pathway hyperactivated. This raises the question of whether the reduction is the consequence of a non-cell autonomous effect the previously described disorganization of the CNS. The experiment on figure S9 suffers from the same shortcomings. In addition, using GFP as a proxy for Fas2 expression is flawed due to the different subcellular localization and levels of expression/staining. A comparison of panels A and B (1) with the respective GFP mask shows a clear overexposure of the GFP mask broadening the original signal several axon diameters.

The choice of stage 16 embryos to analyze expression of Nr4 IV (fig 5) suffers from the same concerns as previously mentioned.

Reviewer #3 (Remarks to the Author):

This manuscript by Karkali et al investigates the involvement of JNK signalling in nervous system morphogenesis in *Drosophila* embryos. They propose a network involving various genes that ultimately regulate cell adhesion and tension, to influence neuropile structure within each segment and ventral nerve cord condensation. The manuscript has abundant data focusing on the effect of loss or gain of function of multiple genes. However, the overall message as well as some specific aspects are unclear and lacking in focus in parts. Here are some specific suggestions to improve the manuscript:

1. Pioneer neurons. This nomenclature is not accurately used, and the authors mix motoneurons and pioneer neurons of longitudinal connectives. The pioneer neurons of the longitudinal connectives are dMP2, vMP2, MP1 and pCC; the pioneer of the motoneurons is aCC. RP2 and VUMs are motoneurons, but what is the experimental evidence that they are pioneer motoneurons? Of the longitudinal connectives, there is no mention of vMP2 nor MP1, so evidence is lacking that all pioneer neurons are involved in the process under investigation. The authors should cite reference describing pioneer neurons (eg by Hidalgo for longitudinal pioneer neurons, and by Sanchez-Soriano for motoneuron pioneering). It had been described that ablation of all pioneer neurons is required to cause major disruption to the neuropile (Hidalgo and Brand 1997). Hence, it seems unlikely that interference with JNK signalling in only dMP2 and pCC would be sufficient to affect the longitudinal connectives, and perhaps other cells are involved in their phenotypes. Since FasII is expressed in all pioneer neurons, the authors could use FasII to determine whether their JNK network operates in all of them (eg testing co-localisation with *puc*>GFP as in Figure 2), not just pCC and dMP2. Title and text should be modified accordingly.

2. Gene network. Figure 7 shows inferences based on LOF and GOF phenotypes. However, to order the network (ie place the direction of arrows) this kind of work requires epistasis and rescue analysis (eg GOF for one gene in the LOF genotype for another). (Only one rescue experiment was carried out, and it yielded a negative result). Furthermore, the diagram in Figure 7 does not explain some of the phenotypes observed (eg *puc* decrease with both JNK LOF and GOF).

3. Solving these relationships is important because others have shown that interference with glia or macrophages prevent VNC condensation. So their phenotypes could also be due to altering JNK signalling in these cells. Furthermore, JNK could also be affecting changes in cell shape which would affect VNC condensation independently of axonogenesis.

Specific points:

4. Figure 2, FigS5 and S6: separate channels must be shown for colocalization evidence
5. Figure 3: lacks a control, quantification and statistical analyses.
6. All figures with stats: information on stats test, multiple comparison corrections and sample sizes are missing and must be provided.
7. VNC is analysed only in some cases. Consistency in the analysis would improve the work, particularly if claims are to be made about the influence on VNC condensation.
8. Discussion is too long and baroque in places, obscuring the meaning of the work.

REVIEWERS COMMENTS

REPLIES IN BLUE

Reviewer #1 (Remarks to the Author):

Since I am not a neurobiology expert, when reviewing this manuscript the editor asked me to look specifically at the genetic aspects of the work; the nature of the different mutations and the utilization of the Gal4/UAS system to manipulate the expression of the JNK pathway.

The experimental approach of the work is simple and straightforward: having described the overall structure of the ventral nerve cord (VNC) in *Drosophila* embryos, the authors describe how alterations of JNK function impinge on the architecture of the VNC.

To manipulate JNK activity the authors make extensive use of mutations at the *puc* (puckered) gene, a JNK target that functions as a negative regulator of the pathway, as previously shown by the senior author of the work. Thus *puc* mutations cause inappropriate elevation of JNK activity. The authors find that in *puc* mutant embryos the condensation of the VNC is abnormal. Moreover, the authors use a *puc*-Gal4 line to locate JNK expression in specific groups of neurons of the VNC. To establish that the alterations found in the VNC of *puc* mutants are due to excessive JNK activity and not to general disturbance of the mutant embryos, the authors use several Gal4 lines to force high JNK expression (using constitutive active form of the Jun kinase Hemipterous) in the subsets of neurons that normally express *puc*. They find alterations in the VNC consistent with those observed in *puc* mutants. Interestingly, suppression of JNK activity (using a dominant negative form of the kinase Basket) in the same neurons cause similar morphological defects in the VNC. The authors also make use of *puc* mutations to identify downstream factors that may be associated with the VNC phenotypical alterations. The levels of Fasciclin 2 (*Fas 2*), a cell adhesion molecule, appear to be reduced. Moreover, *Fas 2* mutations affect VNC condensation, suggesting that *Fas 2* mediates JNK activity on VNC condensation. They also find other factors like neurexin and *zfh1*, whose expression is abnormal in *puc* mutants and their loss of function causes defects in VNC architecture.

In sum, the authors make effective use of mutations affecting JNK activity and of Gal4/UAS mediated manipulations to describe the requirements for JNK function in the formation of the *Drosophila* Ventral Nerve Cord. Essentially it is descriptive work, supported by some nice figures and movies; some genetic elements have been identified, but clearly the picture is far

from complete. How much advance it represents in the current understanding of the Drosophila Nervous System I cannot tell

We thank the reviewer for the accurate overview of the events we describe. It is reassuring to see that we have managed to describe our findings in an understandable way.

Regarding significance, we believe this article provides important insights into the signaling mechanisms that orchestrate central nervous system morphogenesis, and that translate cells communication into tissue's structural organization refinement. Historically, these issues have been very difficult to tackle and so far they have been barely explored.

Reviewer #2 (Remarks to the Author):

In this manuscript Karkali et al use spatial 3D correlation analysis to detect conserved domains within Fas2 expressing longitudinal axon bundles which they define as structural nodes. The distance among these structural nodes is used to determine defects in ventral nerve cord architecture in different genetic backgrounds. The authors analyze *puc* Loss-of-function, a background where the JNK pathway is activated as well as well as a dominant negative Bsk or constitutively active Hep driven in small subsets of neurons. They conclude that the JNK pathway is required for the structural organization of the ventral nerve cord. They further analyze Fas2 expression on those backgrounds to conclude that *fas2* is downstream the JNK pathway, although not the only player. Finally, they show how *Zfh1* mediates the expression of Fas2 and *Nrx IV* in opposite ways downstream the JNK pathway. The proposed model where JNK regulates *Zfh1*, which in turn downregulates *Nrx IV* and upregulates Fas2 generating an autoregulatory loop at the level of JNK to regulate VNC development could be interesting to a wide audience. Unfortunately, the evidence provided and methodological problems make the author's conclusions premature.

We thank the reviewer for his/her appreciation of the interest our manuscript could have to a wide audience. Regarding the evidence and methodological problems raised, we tried to make our conclusions more solid with new experiments, following Reviewer's guidance.

Fig1 -3D correlation analysis:

A better description of the methodology and panels in figure 1 is required. There are axes in the graphs without labels There should be a color-coded scale for the correlation. Do the double

headed arrows actually point to the position in panel A?. There seems to be a problem with the scale bar as the whole neuropil seems to be 60 μ m while panel B, the cross-correlation matrix is 50 bin (1.65 μ m/bin 82.5 μ m total), where the image is cropped to the VNC. It is not clear what does this refer to. Are the bins 1.65 μ m or 1.65 μ m²?. In addition the cross-correlation matrixes seem to correspond to hemisegments as the repeated correlation hotspots seem to correspond to Fas2 fascicles on one half of the embryo. This is not specified anywhere.

On the amended Figure 1, we have incorporated all changes suggested. Better explanations in the text (**lines 78 to 96 and 326 to 335**) and legend (**lines 780 to 805**) are also provided.

Fig3 Targeted activation and inhibition of the JNK pathway:

In this experiment the authors inhibit or hyperactivate the JNK pathway in subsets of neurons. Satisfyingly, when the pathway is tampered with on U motor neurons there are no overt defects in longitudinal Fas2 pathways. However, when the JKN pathway is hyperactivated or inhibited with the RN2-Gal4 driver there seem to be major defects in the VNC. This is quite significant. The RN2-Gal4 driver has been extensively characterized and it drives expression in only 3 neurons, the aCC and RP2 motor neurons that pioneer the ISN and pCC. Such an effect on Fas2 longitudinal fascicles of a single neuron are quite significant and would point towards an instructive role of pCC, that has never been described.

We agree with the reviewer in the significance of the instructive role of the pCC. Indeed, we felt surprised during our early analyses. We think our data strongly support an early instructive "structural" role for the pCC, probably in coordination with other instructive elements. Among these, we cannot exclude, *prima facie*, the contribution of the aCC and RP2 neurons. They may indirectly affect the organization of the neuromere around the 3D structural nodes. The aCC and pCC cell bodies are located in proximity to the nodes and positional and morphological alterations in the RP2 motoneuron pioneers could affect adjacent, connected interneurons (**Discussion, lines 456 to 467**). In this scenario, the effects observed employing the early RN2 Gal4 line might not be caused just by a single neuron, the pCC, but by a seeding effect of the rest of the RN2 early born neurons.

In other genetic backgrounds (for example *eve* nulls on the RN2 neurons), where this interneuron mis-projects across the midline, longitudinal fascicles are properly fasciculated.

The effect reported in this manuscript (lines 174-178) is quite severe and may point to a non-cell autonomous effect. This major discrepancy should have been at discussed least.

Fujioka et al, 2003, Reference 48, found that abolishing *eve* expression exclusively in the RP2/aCC/pCC results in axonal morphology aberrations and cell body mis-positioning of these neurons, although the longitudinal fascicles seemed to be properly formed. However, no in depth structural analysis was presented and no self cross-correlation with regard to the distribution or integrity of Fas 2 bundles was performed. Indeed, observing the data corresponding to stage 16 embryos in this report, becomes evident that when the RN2 neurons are *eve* null, the scaling of the axonal scaffold (the position of the anterior and posterior commissures and the inter-commissural space) was aberrant and that the VNC condensation was defective. All in all, the phenotype of the *eve* null RN2 neurons is rather similar to that of the RN2>HepCA embryos.

It is also worth mentioning that in *eve* mutant embryos, Bsk mRNA levels are dramatically reduced in the CNS (**Zarin et al, 2014; Reference 40**). Therefore, it is fair to assume that at some extend the phenotypic aberrations observed in the CNS of these embryos are related to JNK signaling down regulation. Quite possibly, HepCA overexpression in RN2 neurons due to the negative feedback loop mediated by Puc could mimic Bsk loss of function conditions. It is still unclear if Eve is upstream of JNK signaling in these neurons.

These issues are now thoroughly discussed in the amended manuscript (**Discussion, lines 359 to 371**).

Figs 4, S8,S9 Fas2 regulation by JNK.

The authors quantify Fas2 fluorescent staining in *puc* null embryos (fig 4) and conclude that there is a reduction. The authors also point out that in *puc* mutants there is also an increase in apoptosis (line 131). Apoptosis is not quantified in the present work but become relevant here as a reduction in Fas2 signal could be due to increase in apoptosis of Fas2+ neurons.

The reviewer is right and initially we could not discard the possibility that the reduction in Fas 2 levels in *puc* null embryos, was due to an increase in apoptosis (Figure S4). Yet, the increase in NrX IV levels in the same conditions (Figure 5) advocates against this scenario. Further, when Hep^{CA} is expressed in RN2 cells, the affected neurons do not die and, overall cell death is

reduced, while Fas 2 is still downregulated. This is now discussed in the manuscript (**lines 229 to 235 and 438 to 446**).

In an attempt to determine Fas2 regulation in a cell specific manner the authors hyperactivate the JNK pathway specifically in RN2 neurons. In these circumstances they observe a significant Fas2 reduction at nodes and internodes at stage 16. This experiment should have been done at stage 12-13 for various reasons. To have a single cell resolution the experiment has to be performed when axons are starting to project and at those stages Fas2 is clearly present. At stage 16, only the pCC axon (within a bundle of several tenths) has the JNK pathway hyperactivated. This raises the question of whether the reduction is the consequence of a non-cell autonomous effect the previously described disorganization of the CNS.

This experiment has been repeated at stage 13 and the results are essentially equivalent to those observed at stage 16: a reduction of Fas 2 levels and a failure of the VNC to condense (**New Figure S8**).

The experiment on figure S9 suffers from the same shortcomings. In addition, using GFP as a proxy for Fas2 expression is flawed due to the different subcellular localization and levels of expression/staining. A comparison of panels A and B (1) with the respective GFP mask shows a clear overexposure of the GFP mask broadening the original signal several axon diameters.

The GFP in Figure S9 (**now Figure S10**) displays RN2 expression and not Fas2. It is employed to distinguish (as a mask) the territory allocated to the GFP expressing cells allowing to calculate which proportion of Fas 2 expression (detected with an antibody) associates to RN2 positive cells. In the collected images, some variable faint GFP signal was at the edge of the dynamic range and could not be valued leading to under estimated figures. To overcome this problem, we overexposed the signal, creating masks maximizing GFP levels. In this way, we categorized Fas 2 expression as GFP positive or negative including regions with low RN2 expression levels. We believe that with our approach, we got closer to real quantitative measurements.

The choice of stage 16 embryos to analyze expression of NrX IV (fig 5) suffers from the same concerns as previously mentioned.

Checking the expression of Nr_x IV at early time points after HepCA overexpression in RN2-Gal4 was extremely difficult as its expression level was much lower than at stage 16/17, when is, indeed, very low. However, we managed to retrieve some data (**New Figure S8**) and found, that contrary to stage 16, upon JNK hyperactivation, Nr_x IV expression was downregulated. Thus, Nr_x expression experiences a dynamic temporal regulation that could be justified by a shift in the balance (JNK inhibits Zfh1 expression and Zfh1 reduces JNK phosphorylation in a mutual negative loop) between the positive input from Bsk and the negative input from Zfh1 on Nr_x IV expression. The exploration of this potential mechanism remains to be addressed. Yet, we consider it represents a minor point in the current manuscript.

Reviewer #3 (Remarks to the Author):

This manuscript by Karkali et al investigates the involvement of JNK signalling in nervous system morphogenesis in *Drosophila* embryos. They propose a network involving various genes that ultimately regulate cell adhesion and tension, to influence neuropile structure within each segment and ventral nerve cord condensation. The manuscript has abundant data focusing on the effect of loss or gain of function of multiple genes. However, the overall message as well as some specific aspects are unclear and lacking in focus in parts.

We have now rewritten substantial parts of the manuscript and tried to make our conclusions clearer. Adding the new information asked for by the reviewers demanded to expand the text and several new panels have been added to multiple figures. Yet, we made an effort to be more focused and to define more precisely the significance of the study.

Here are some specific suggestions to improve the manuscript:

1. Pioneer neurons. This nomenclature is not accurately used, and the authors mix motoneurons and pioneer neurons of longitudinal connectives. The pioneer neurons of the longitudinal connectives are dMP2, vMP2, MP1 and pCC; the pioneer of the motoneurons is aCC. RP2 and VUMs are motoneurons, but what is the experimental evidence that they are pioneer motoneurons?

The nomenclature of the neurons has been updated all throughout the manuscript following reviewer's guidance. Regarding RP2 and VUM motoneurons, they were described as motoneuron pioneers of the ISN by **Sanchez-Soriano and Prokop, 2005, Reference 22**.

Of the longitudinal connectives, there is no mention of vMP2 nor MP1, so evidence is lacking that all pioneer neurons are involved in the process under investigation. The authors should cite reference describing pioneer neurons (eg by Hidalgo for longitudinal pioneer neurons, and by Sanchez-Soriano for motoneuron pioneering).

We have found (see the new data in **Figure 2H** - stage 13) that vMP2 and MP1 do not express *puc*. We have not studied them further. The manuscript' title has been changed in agreement and we now refer to a subset of pioneers, as those involved in modulating the architectural organization of the VNC in response to JNK. If other pioneers, *puc* negative, also participate in modulating the VNC structural organization and its condensation remains to be investigated.

The two manuscripts referred by the reviewer are now properly cited in the new version (**References 22 and 23**).

It had been described that ablation of all pioneer neurons is required to cause major disruption to the neuropile (Hidalgo and Brand 1997). Hence, it seems unlikely that interference with JNK signalling in only dMP2 and pCC would be sufficient to affect the longitudinal connectives, and perhaps other cells are involved in their phenotypes.

It is well known that killing subsets of pioneer neurons does not deeply affect Fas 2 longitudinals and it is necessary to kill all pioneer neurons to provoke major disruptions (**Hidalgo and Brand, 1997, Reference 23**). However, by interfering in the activity of the JNK cascade in a relevant subset of pioneers, we are generating novel phenotypes that stand apart from the role of these neurons in axon guidance, and are related to something different, the 3D structural organization of the neuropile. This is now discussed in the manuscript (**lines 229 to 235 and 438 to 446**)

We must also consider that there are qualitative differences that distinguish between pioneer neurons death, and limiting their capacity to interpret and integrate complex cues in a developing environment by altering the JNK pathway activity.

Employing the very restricted *RN2-Gal4* and *MzVum-Gal4* to interfere in JNK signaling, unambiguously shows that these cells (under the control of the JNK pathway) are essential for the 3D structural organization of the VNC. Yet, we are quite certain that other neurons (maybe JNK negative), not studied in this work could also have an important role in this process.

We have rephrased our conclusions and made them less categoric, acknowledging the potential participation of other partners (**lines 456 to 467**).

Since FasII is expressed in all pioneer neurons, the authors could use FasII to determine whether their JNK network operates in all of them (eg testing co-localisation with *puc*>GFP as in Figure 2), not just pCC and dMP2.

This experiment was performed and is now included in the manuscript (**Figure 2H**). Fas2 expression was studied in stage 13 embryos and colocalization with *puc*>GFP was examined. *puc* seems to be excluded of the vMP2 and MP1 pioneers. The new data has been added to the text (**lines 153 to 155**).

Title and text should be modified accordingly.

The title has been modified as requested and the main text has been updated following reviewer's recommendations.

2. Gene network. Figure 7 shows inferences based on LOF and GOF phenotypes. However, to order the network (ie place the direction of arrows) this kind of work requires epistasis and rescue analysis (eg GOF for one gene in the LOF genotype for another). (Only one rescue experiment was carried out, and it yielded a negative result).

The displayed network aims to describe the links between its different elements in terms of expression control, at the protein level (Zfh1, Fas 2 and NrX IV), or activity (JNK phosphorylation). We have clarified this in the new version of Figure 7 (different graphic representation of expression and activation links is used). The model helps to show how by disturbing the activity of the JNK pathway, and through intermediate elements such as Zfh1, Fas2 and NrX IV, the condensation and architectural organization of the VNC are affected. Clearly, we have not saturated the system and, although all the identified partners alter the condensation and organization of the VNC axonal network, as the JNK pathway does, other unknown elements may also be at work.

In this scenario, Fas 2 rescue of JNK loss of function failed, as, most probably, any other rescue attempt will do. It is important to consider that the robustness of the 3D architectural

organization of the VNC relies on the activity of the JNK pathway acting on multiple adhesion and guidance elements, that they work coordinately and that multiple negative and positive regulatory loops (those already identified and probably others) also intervene.

These considerations are now incorporated in the text (**Discussion, lines 404 to 417**).

Furthermore, the diagram in Figure 7 does not explain some of the phenotypes observed (eg *puc* decrease with both JNK LOF and GOF).

Single negative feedback loops are involved in four distinct signaling functions: basal homeostasis, output limitation, adaptation, and transitory control. The effect of JNK LOF on *puc* expression is self-explanatory as *puc* is an early expressing gene under the control of the AP1 complex. The JNK GOF effect on *puc* expression, on the other hand, will be the result of the strong (transient) early inhibitory effect of *puc* on JNK activity, which will, in the long term inhibit its own expression. This loop has been previously described in other morphogenetic processes in *Drosophila* such dorsal closure and border cell migration, as well as in the imaginal discs wound healing response. Our data suggest that this negative regulatory loop is at work on the young CNS.

These considerations are now incorporated in the text (**lines 200 to 207**).

3. Solving these relationships is important because others have shown that interference with glia or macrophages prevent VNC condensation. So their phenotypes could also be due to altering JNK signalling in these cells.

Glia migration as well as macrophages dependent cell body clearance and ECM deposition have been reported to alter condensation. Indeed, we have shown that glia is mechanically important for the condensation process (**Karkali et al, 2022, Reference 9**).

puc is expressed in VNC neurons and also in macrophages but not in glia. Our interference experiments employing the MzVum and RN2 drivers, that are not expressed in macrophages, indicates that the function of the JNK pathway in the structural organization of the VNC seems to be mediated just by this subset of pioneer neurons (**Discussion, lines 447 to 455**). Testing if the JNK pathway is also important for this process in macrophages is under current investigation, but we think it is out of the scope of this manuscript.

Furthermore, JNK could also be affecting changes in cell shape which would affect VNC condensation independently of axonogenesis.

This is indeed very relevant. The packing of neurons will probably be affected by shape changes in their cell bodies. Yet, to monitor 3D shape changes in neuronal cell bodies within the nerve cord is essentially impossible due to their dense packing. To overcome this issue, in a previous work (**Karkali et al, 2022, Reference 9**), we resorted to measure cell bodies density. Cell bodies compact on a stereotyped way, suggesting that cells intercalation plasticity is a necessary requisite for compaction. Therefore, given that in JNK hyperactivation conditions, VNC condensation does not progress it is safe to assume that neurons shape and packing are most probably affected (**Discussion, lines 418 to 424**).

Specific points:

4. Figure 2, FigS5 and S6: separate channels must be shown for colocalization evidence

We think that to display separate channels will not help as evidence for colocalization. The colocalization (and lack of colocalization) of signals in these figures is clear enough and we deem showing separate channels unnecessary. It would increase the number of panels in the figures to an unmanageable level.

5. Figure 3: lacks a control, quantification and statistical analyses.

Figure 3 control is in Figure 1. All analyzed individuals for each condition are represented in the graphs. Sample sizes are now provided.

6. All figures with stats: information on stats test, multiple comparison corrections and sample sizes are missing and must be provided.

All stats have been updated throughout the manuscript.

7. VNC is analysed only in some cases. Consistency in the analysis would improve the work, particularly if claims are to be made about the influence on VNC condensation.

New VNC length measurements have been included for all the genetic conditions analyzed.

8. Discussion is too long and baroque in places, obscuring the meaning of the work.

Although the amount of data has increased from the previous version, we tried to simplify the discussion eliminating non-essential considerations. We aimed to point to the critical take home messages derived from our data and analyses.

REVIEWERS' COMMENTS

Reviewer #2 (Remarks to the Author):

On their revised manuscript Karkali et. al have properly addressed some of my comments Figure 1 is clearly much better explained within the text and in the legend. The specific role of pCC or is discussed at length. The authors have made an attempt to show changing levels of NrX IV or Fas2 with single cell resolution in a new figure but these results are not convincing as the fluorescent signal seems saturated or is a poor staining. My perception is still that the work is still preliminary and major conclusions are primarily based on expression levels of different molecules based on fluorescent intensity. A more thorough analysis where more lines of evidence were provided, for example genetic interactions and/or other complementary approaches with in vitro experiments are required.

Reviewer #3 (Remarks to the Author):

On the whole, the authors have made significant improvements to the manuscript and I am satisfied with the way they have responded to the criticisms I raised.

Just a few rather minor improvements would help improve the quality of the work:

In the abstract “the fasciculation of axons of pioneer neurons” is unclear. The pioneer neurons are the first ones to lay down axonal tracks, and follower neurons fasciculate along them. FasII labels around 30% of neurons. I recommend they change the expression to “the fasciculation along pioneer axons”.

In the introduction, the sentence line 53-54 “Indeed, the JNK recently emerged as a regulator for axonal growth and regeneration...” drifts away from the focus of this work and does not make a relevant point. It is well known that many genes have pleiotropic functions, but this does not mean that such functions are related.

There are various “the” that are not correct and should be removed: e.g. above “the JNK recently emerged...”; line 106 “the JNK activity”... Check for others.

Line 111-112 starting: “We found that....” Where are these data? Figure must be cited.

Figure 2H is good, but it does not indicate where vMP2 and MP1 are, and this is required in order to claim that puc is not there. So the authors must label the positions of vMP2 and MP1 in the figure.

Figure 3: the BskDN and HepCA phenotypes still have not been compared to the control. Although the data for the control are given in Figure 1, there is no comparison of the BskDN and HepCA phenotypes relative to the control. Since they have the quantifications, these data could be converted to a graph to compare these genotypes to control and analyse with statistics. Is this not possible with the data they

already have?

The data on the statistical analysis is still incomplete. They only provide the sample size and the p values. In the figure legends, the authors need to provide: the sample sizes; indicate the statistical tests used when comparing two genotypes (e.g. for quantitative data, parametric Student t-test or non-parametric Mann Whitney U test); and if parametric, indicate whether the bar charts indicate the means and what the error bars represent, best if they are s.d.; when comparing three genotypes or more, they must provide the statistical test for the group analysis (e.g. One Way ANOVA if parametric or Kruskal Wallis ANOVA if not parametric) and the multiple comparisons corrections or post-hoc tests (e.g. post-hoc Dunnett for parametric comparisons to a fixed control, post-hoc Dunn for non-parametric comparisons to a fixed control, Boneferroni everything against everything, etc); and they must provide the p values for each of these (ie for the comparisons of 2 sample types, for the group analysis and for the post-hoc comparisons). The convention is that the asterisks refer to the post-doc multiple comparisons corrections. The authors have done all these analyses, but have not provided the evidence and it is common to provide all of these details in the figure legends.

Claims of colocalization are not supported unless separate channels are shown. They can show separate channels within the supplementary figures. Authors can opt for showing the evidence or removing the claims.

REVIEWERS' COMMENTS

Reviewer #2 (Remarks to the Author):

On their revised manuscript Karkali et. al have properly addressed some of my comments Figure 1 is clearly much better explained within the text and in the legend. The specific role of pCC or is discussed at length. The authors have made an attempt to show changing levels of Nr4-1 or Fas2 with single cell resolution in a new figure but these results are not convincing as the fluorescent signal seems saturated or is a poor staining. My perception is still that the work is still preliminary and major conclusions are primarily based on expression levels of different molecules based on fluorescent intensity. A more thorough analysis where more lines of evidence were provided, for example genetic interactions and/or other complementary approaches with in vitro experiments are required.

Nr4-1 expression is, indeed, very low at the stages we are studying. Signal is much more evident at larval stages. Yet, we have managed to retrieve proper and reproducible quantitative and qualitative information as described in the manuscript.

While more lines of evidence could be provided, to add in vitro experiments or more genetic tests will not alter the main message of the manuscript. Molecular mechanisms must be explored further but this is out of the scope of this work.

Reviewer #3 (Remarks to the Author):

On the whole, the authors have made significant improvements to the manuscript and I am satisfied with the way they have responded to the criticisms I raised.

Just a few rather minor improvements would help improve the quality of the work:

In the abstract “the fasciculation of axons of pioneer neurons” is unclear. The pioneer neurons are the first ones to lay down axonal tracks, and follower neurons fasciculate along them. FasII labels around 30% of neurons. I recommend they change the expression to “the fasciculation along pioneer axons”.

Following the reviewer' suggestion we changed this sentence as suggested.

In the introduction, the sentence line 53-54 “Indeed, the JNK recently emerged as a regulator for axonal growth and regeneration...” drifts away from the focus of this work and does not make a relevant point. It is well known that many genes have pleiotropic functions, but this does not mean that such functions are related.

We have modified this sentence in the introduction.

There are various “the” that are not correct and should be removed: e.g. above “the JNK recently emerged...”; line 106 “the JNK activity”... Check for others.

We already did a full recheck of the text to find and correct grammar mistakes and redundancies.

Line 111-112 starting: “We found that...” Where are these data? Figure must be cited.

The figure is now cited.

Figure 2H is good, but it does not indicate where vMP2 and MP1 are, and this is required in order to claim that puc is not there. So the authors must label the positions of vMP2 and MP1 in the figure.

In this figure we could denote the position of the vMP2 and MP1, but with low accuracy, as these neurons are not labelled. We prefer to not include this information to avoid potentially misleading the reader.

Figure 3: the BskDN and HepCA phenotypes still have not been compared to the control. Although the data for the control are given in Figure 1, there is no comparison of the BskDN and HepCA phenotypes relative to the control. Since they have the quantifications, these data could be converted to a graph to compare these genotypes to control and analyse with statistics. Is this not possible with the data they already have?

The comparison was done at a descriptive level and discussed in the text, with the key findings being the increment in intermodal distance and the loss of the correlation A/P asymmetry. The Reviewer is right we could represent these data in a graph but this would not facilitate the comparison as nodes spacing is variable in the different genotypes and correlations tend to become noisy. We strongly believe that the opted data representation highlights the qualitative differences

scored in a simple and direct way. Although we could generate further statistical tests we think they are not necessary.

The data on the statistical analysis is still incomplete. They only provide the sample size and the p values. In the figure legends, the authors need to provide: the sample sizes; indicate the statistical tests used when comparing two genotypes (e.g. for quantitative data, parametric Student t-test or non-parametric Mann Whitney U test); and if parametric, indicate whether the bar charts indicate the means and what the error bars represent, best if they are s.d.; when comparing three genotypes or more, they must provide the statistical test for the group analysis (e.g. One Way ANOVA if parametric or Kruskal Wallis ANOVA if not parametric) and the multiple comparisons corrections or post-hoc tests (e.g. post-hoc Dunnett for parametric comparisons to a fixed control, post-hoc Dunn for non-parametric comparisons to a fixed control, Bonferroni everything against everything, etc); and they must provide the p values for each of these (ie for the comparisons of 2 sample types, for the group analysis and for the post-hoc comparisons). The convention is that the asterisks refer to the post-doc multiple comparisons corrections. The authors have done all these analyses, but have not provided the evidence and it is common to provide all of these details in the figure legends.

All the Figures legends have been modified and the statistical tests clarified. Further, all individual data points have been now added to the graphic representations.

Claims of colocalization are not supported unless separate channels are shown. They can show separate channels within the supplementary figures. Authors can opt for showing the evidence or removing the claims.

We think separating the channels in all figures containing double stainings would add to the paper's volume and would confuse the readers, without contributing to either data interpretation or to the message of the paper.